



# A dilatant visco-elasto-viscoplasticity model with globally continuous tensile cap: stable two-field mixed formulation

Anton A. Popov[1], Nicolas Berlie[1], and Boris J. P. Kaus[1]

[1]Institute of Geosciences, Johannes Gutenberg University, Mainz, Germany

**Correspondence:** Anton A. Popov (popov@uni-mainz.de)

**Abstract.** Rocks break if shear stresses exceed their strength. It is therefore important for typical geoscientific applications to take shear failure mechanism and the subsequent development of mode-II shear bands or faults into account. Many existing codes incorporate non-associated Drucker-Prager or Mohr-Coulomb plasticity models to simulate this behavior. Yet, when effective mean stress becomes extensional, for example when fluid pressure becomes large, the dominant failure mode changes

to a mode-I (opening) mode, which initiates plastic volumetric deformation. It is rather difficult to represent both failure modes in numerical models in a self-consistent manner, while also accounting for the nonlinear visco-elastic host rock rheology, which varies from being nearly incompressible in the mantle to being compressible in surface-near regions. Here, we present a simple plasticity model that is designed to overcome these difficulties. We employ a combination of a linearized Drucker-Prager shear failure envelope with a circular tensile cap function in way that ensures continuity and smoothness of both yield surface and flow

potential in the entire stress space. A Perzyna-type viscoplastic regularization ensures that the resulting localization zones are mesh-insensitive. To deal with the near incompressibility condition, a mixed two-field finite element formulation is employed. The local nonlinear iterations at the integration-point level are used to determine the stress increments. The global Newton-Raphson iterations are applied to solve the discretized momentum and continuity residual equations. The presented plasticity model is implemented in an open-source 2D unstructured finite element code GeoTech2D. The results of several typical test

cases that range from crustal scale deformation to the propagation of fluid-induced tensile failure zones demonstrate rapid convergence. The robustness of the solution scheme is enhanced by the adaptive time stepping algorithm.

## 1 Introduction

Plastic deformation in brittle rocks manifests itself in two ways: as mode-II shear faults, or as mode-I failure zones, which results in an opening of the rock in a crack-like manner and therefore produces volumetric strains. Having the ability to

simulate both faulting modes numerically in a self-consistent manner is important to simulate near-surface brittle deformation, or simulate cases where the fluid pressure is high, such as during the initiation of hydraulic fractures or magma-filled dykes. Most plasticity models for rocks are pressure-dependent. Whereas mode-I plasticity models involve dilation or volumetric deformation, experiments demonstrate that mode-II faults usually involve only a bit of dilation during the initiation stages, but are dilation-free or incompressible after some deformation (e.g., Vermeer and de Borst, 1984). Thus, numerical models



that incorporate general plasticity models for rocks need to have accurate pressure fields, and should be able to deal with both compressible and incompressible deformation.

Quasi-static long-term flow of viscous materials, such as hot rocks in the Earth's mantle or rocksalt in the salt diapirs, is nearly incompressible, which is well-known to require a special treatment in numerical models to avoid numerical artifacts in the pressure field (e.g., Zienkiewicz and Taylor, 2000). This is of particular importance if the deformation mechanism is

sensitive to pressure (as is the case for many plasticity models). Within a finite element discretisation framework, a typical approach is to use a so-called mixed formulation which combines velocity (displacement increments) and pressure as primary variables. Yet not all interpolation types can be adopted since they need to satisfy a set of stability criteria commonly known as the LBB conditions (e.g., Gatica, 2014). Examples of stable finite elements types include the Taylor-Hood approximation on quadrilateral and hexagonal shape e.g. $Q_2 \times Q_1$, which uses continuous quadratic and linear polynomials for the velocity and

pressure, respectively. The discontinuous pressure version is typically represented by $Q_2 \times P_{-1}$ elements (see e.g., Thieulot and Bangerth, 2022, for an overview of most commonly used combinations). The latter element is very widely used in geodynamic simulations due to its reliability in dealing with sharp jumps in viscosities (Kronbichler et al., 2012; May et al., 2014; Deubelbeiss and Kaus, 2008).

The simplex element shape can also be used to construct stable discretizations. A particular candidate is the conforming

triangular Crouzeix-Raviart element ($P_2^+ \times P_{-1}$), which uses quadratic interpolation enhanced with a bubble function for the velocities (displacement increments), and a linear discontinuous interpolation for pressure (Crouzeix and Raviart, 1973). The triangular shape of the element facilitates discretizing complex geometrical domains, whereas the discontinuous pressure interpolation enables enforcing mass conservation at the element level and is in general advantageous for difficult problems with abrupt material contrasts (Pelletier et al., 1989). The element has been shown to behave robustly in practice (Dabrowski

et al., 2008), and can be generalized to 3D (Crouzeix and Raviart, 1973).

As an alternative to the stable finite elements, it is also possible to apply a staggered grid finite difference discretization (Harlow and Welch, 1965). This method is computationally cheap and also demonstrates excellent and reliable performance for the relevant problems (Gerya and Yuen, 2007; Tackley, 2008; Kaus et al., 2016; Räss et al., 2017; Deubelbeiss and Kaus, 2008). The staggered finite difference formulation was proven to be a stable formulation for the incompressible (Shin and

Strikwerda, 1997), and compressible Stokes equations (Eymard et al., 2010), and was numerically demonstrated to be LBB-stable (Gerya et al., 2013).

Despite that a number of stabilized equal-order interpolation finite elements have been proposed (e.g. Dohrmann and Bochev, 2004; Cioncolini and Boffi, 2019), their performance does not practically bring a satisfactory level of robustness for the cases of large parameter variation (Thieulot and Bangerth, 2022). Selecting a stable discretization is therefore an important milestone for

successful numerical implementation of the nonlinear rheological models that involve nearly-incompressible material behavior.

Implementing plastic rheologies in the context of a two-field formulation is not straightforward. The problem is caused by the presence of stress-like variables (pressure) among the primary unknowns, whereas stress integration algorithms are typically formulated in a strain-driven fashion (de Borst et al., 2012). Pressure-independent models, such as J2-plasticity (Pastor et al., 1997), or incompressible Drucker-Prager plasticity (Gerya and Yuen, 2007; May et al., 2014; Kaus et al., 2016; Glerum et al.,





2018) poses no additional difficulty because pressure remains unaltered during the local stress update. For nonzero dilatation cases, as well as for plastic compaction or tensile failure modes, this is no longer the case, and therefore at least two potential solutions may be elaborated. The first approach computes deviatoric stresses from the kinematical variables (velocities or displacement increments) using the standard strain-driven algorithm, wheres the globally discretized pressure is treated as an actual spherical part of the Cauchy stress tensor (Commend, 2001; Commend et al., 2004). This type of algorithms can be

referred to as *true pressure formulation*. The second approach treats the globally discretized pressure as a trial visco-elastic pressure (Duretz et al., 2021), which can be termed as *trial pressure formulation*. Additionally, it should be noted that pressure selection problems can be completely avoided by adopting a mixed formulation that involves strains as global variables (e.g., Benedetti et al., 2014). This approach is computationally much more expensive and therefore beyond the scope of this paper.

It is often quite difficult to describe different plastic failure regimes, such as plastic compaction, or tensile failure (mode I)

and shear failure (mode II) with a single yield surface. A standard approach is therefore to tackle this problem with a multi-surface plasticity model. A two- or three-invariant shear failure envelope is typically combined with compression and tensile caps (e.g., Sandler and Rubin, 1979). A non-smooth intersection of the segments results in the complex algorithmic treatment of the corner regions in the stress space (e.g., Simo et al., 1988). To avoid this difficulty, alternative smooth multi-surface models have been proposed where the focus was mostly placed on enforcing continuity and differentiability of the yield surface (Swan

and Seo, 2000; Dolarevic and Ibrahimbegovic, 2007; Motamedi and Foster, 2015). This, however, does not automatically guarantee continuity in the entire stress space. Plotting isocontours of the yield surface can reveal typical problems that include spurious elastic domains, singularity points, erratic gradients, discontinuities, or non-convex regions (e.g., Stupkiewicz et al., 2014; Golchin et al., 2021). A squared version of the yield functions does not even ensure dimensional consistency between the segments (Swan and Seo, 2000), or may lead to loss of convexity in the stress space (Stupkiewicz et al., 2014). Complex

stress integration procedures are therefore required to determine the set of active yield surface segments (Swan and Seo, 2000; Dolarevic and Ibrahimbegovic, 2007). All these problems lead to severe algorithmic difficulties and convergence problems of local stress update iterations.

To model rock failure, the tensile regime is more important than plastic compaction as it is relevant for modeling dyke propagation (e.g., Rivalta et al., 2015; Li et al., 2023). It is therefore potentially advantageous to describe both failure modes

with a single smooth hyperbolic approximation of the Mohr-Coulomb yield surface (Abbo and Sloan, 1995). This approach, however, completely fails when dilatation angles go to zero, since the plastic flow potential becomes pressure-insensitive in the tensile mode. Carol et al. (1997) proposed an ad-hoc approach to restore the proper tensile behavior by introducing a pressure-dependent dilation coefficient, however, at the cost of destroying the convexity of the flow potential, which can be easily revealed by plotting its isocontours. Li et al. (2023) adopted the same approximation, but in their formulation, the amount

of dilatation in flow potential is dependent on porosity rather than on pressure, which lacks physical explanation. Other existing implementations of tensile failure of rocks in the geodynamic community (e.g., Rozhko et al., 2007; Keller et al., 2013; Kiss et al., 2023) employ non-smooth yield surfaces, and are therefore not guaranteed to produce well-defined stress integration algorithms at least in the context of an implicit time integration scheme.





Classical rate-independent plasticity models that incorporate strain softening, which is typically required by constitutive models of rocks and soils (e.g., Read and Hegemier, 1984), are mathematically ill-posed. This results in the lack of an intrinsic length-scale, severe mesh-dependence and convergence issues (de Borst and Mühlhaus, 1992; Spiegelman et al., 2016). There are several approaches to remediate this problem (e.g. Duretz et al., 2023), with the easiest one being a viscoplastic rate-dependent regularization (de Borst and Duretz, 2020). The latter assumes different ways to formulate the constitutive equation for the viscoplastic strain rate. Namely, Perzyna model (Perzyna, 1966) assumes an explicit equation, Duvaut-Lions model (Duvaut and Lions, 1972) is built upon a relatively simple generalization of a rate-independent plasticity, whereas consistency model (Wang et al., 1997; Heeres et al., 2002) introduces the concept of a rate-dependent yield function. Irrespective of the employed viscoplastic regularization, either Perzyna (e.g., Jacquey and Cacace, 2020a, b), Duvaut-Lions (e.g., Kiss et al., 2023), or the consistency model (e.g., Duretz et al., 2019; Li et al., 2023), a successful resolution of the mesh issue simultaneously with the improvement of the global convergence or stability of the time integration scheme can be achieved.

Building up on previous work, we present a visco-elasto-viscoplastic rheological model that combines linear elastic response with diffusion and dislocation viscous creep mechanisms and a relatively simple Perzyna-type viscoplastic model. The yield surface is composed of a linear Drucker-Prager shear failure envelop and a circular tensile cap in a way that ensures dimensional consistency, convexity, continuity and differentiability throughout the entire stress space. The model allows an arbitrary amount of dilatation in the shear failure regime, without compromising the description of tensile failure. The viscoplastic regularization enables the incorporation of strain softening, such that spurious mesh dependence is avoided and global convergence is ensured. We provide algorithmic details necessary to compute the stresses at the integration points and to solve the resulting global system of nonlinear equilibrium equations with a Newton-Raphson method, including the tangent matrix derivation. The algorithms are implemented in a 2D finite element code that employs stable conforming Crouzeix-Raviart elements, along with incremental displacements and incremental trial pressure as the primary unknowns. A number of examples are given to demonstrate the numerical robustness of the code. The plasticity model presented in this paper can be readily implemented in any finite element or finite difference code that uses a stable mixed two-field formulation.

## 2 Physical model

### 2.1 Volumetric-deviatoric decomposition

Throughout this paper, we assume a standard decomposition of an arbitrary tensor (e.g. $a_{ij}$) into its volumetric (spherical) and deviatoric projections, which are given by, respectively, in the index notation (Einstein summation convention is implied):

$$\text{tr}\,(a_{ij}) = a_{kk}, \quad \text{dev}\,(a_{ij}) = a_{ij} - \frac{1}{3}\,a_{kk}\delta_{ij}, \tag{1}$$

where $\delta_{ij}$ is the Kronecker delta or second order unit tensor: $\delta_{ij} = 1$, for $i = j$, $\delta_{ij} = 0$, for $i \neq j$. For an arbitrary deviatoric tensor we additionally introduce an effective scalar measure equal to the square root of its second invariant, defined as:

$$a_{\text{II}} = \left(\frac{1}{2}a_{ij}a_{ij}\right)^{\frac{1}{2}}. \tag{2}$$



For the Cauchy stress tensor ($\sigma_{ij}$) the volumetric-deviatoric decomposition into stress deviator ($\tau_{ij}$) and mean stress or pressure ($p$), which is assumed to be positive in compression, is defined as:

$$\sigma_{ij} = \tau_{ij} - p\,\delta_{ij}, \quad \tau_{ij} = \mathrm{dev}\,(\sigma_{ij}), \quad p = -\frac{1}{3}\mathrm{tr}\,(\sigma_{ij}). \tag{3}$$

Equivalently, the strain rate tensor ($\dot{\epsilon}_{ij}$) can be decomposed into a strain rate deviator ($\dot{\varepsilon}_{ij}$) and a volumetric strain rate ($\dot{\theta}$) by:

$$\dot{\epsilon}_{ij} = \frac{1}{2}\left(\frac{\partial v_i}{\partial x_j} + \frac{\partial v_j}{\partial x_i}\right) = \dot{\varepsilon}_{ij} + \frac{1}{3}\dot{\theta}\,\delta_{ij}, \quad \dot{\varepsilon}_{ij} = \mathrm{dev}\,(\dot{\epsilon}_{ij}), \quad \dot{\theta} = \mathrm{tr}\,(\dot{\epsilon}_{ij}), \tag{4}$$

where $x_i$, $i = 1, 2, 3$ denote Cartesian coordinates, and $v_i$ are the components of the spatial velocity vector.

## 2.2 Constitutive and conservation equations

The constitutive equations for the deviatoric stress assumes additive decomposition of the deviatoric strain rate into elastic, diffusion creep, dislocation creep and viscoplastic components, respectively, as follows:

$$\dot{\varepsilon}_{ij} = \dot{\varepsilon}_{ij}^{\mathrm{el}} + \dot{\varepsilon}_{ij}^{\mathrm{dif}} + \dot{\varepsilon}_{ij}^{\mathrm{dis}} + \dot{\varepsilon}_{ij}^{\mathrm{vp}} = \frac{\overset{\diamond}{\tau}_{ij}}{2G} + A_{\mathrm{D}}\,\tau_{ij} + A_{\mathrm{N}}\,(\tau_{\mathrm{II}})^{n-1}\,\tau_{ij} + \dot{\lambda}\,\mathrm{dev}\left(\frac{\partial Q}{\partial \sigma_{ij}}\right). \tag{5}$$

Here $G$ denotes the elastic shear modulus, $A_D$ and $A_N$ are the diffusion and dislocation creep prefactors, respectively, $n$ is the power-law exponent, $Q$ is the plastic flow potential function, and $\dot{\lambda}$ is the magnitude of the viscoplastic strain rate (viscoplastic multiplier), which can be explicitly formulated for the Perzyna model (see Sect. 2.3 and 2.4). The Jaumann objective stress rate is defined as:

$$\overset{\diamond}{\tau}_{ij} = \frac{\partial \tau_{ij}}{\partial t} + \tau_{ik}\omega_{kj} - \omega_{ik}\tau_{kj}, \tag{6}$$

where $\dot{\omega}_{ij}$ is the spin tensor, given by:

$$\dot{\omega}_{ij} = \frac{1}{2}\left(\frac{\partial v_i}{\partial x_j} - \frac{\partial v_j}{\partial x_i}\right). \tag{7}$$

The temperature-dependent creep prefactors are:

$$A_{\mathrm{D}} = B_{\mathrm{D}}\,\exp\left[-\frac{E_{\mathrm{L}}}{RT}\right], \quad A_{\mathrm{N}} = B_{\mathrm{N}}\,\exp\left[-\frac{E_{\mathrm{N}}}{RT}\right], \tag{8}$$

where $B_{\mathrm{D}}$, $B_{\mathrm{N}}$ and $E_{\mathrm{D}}$, $E_{\mathrm{N}}$ denote the corresponding constants and activation enthalpies of the diffusion and dislocation creep, respectively, $T$ is the temperature, and $R$ the gas constant.

The volumetric constitutive equation, which can also be termed as continuity equation, is defined similarly:

$$\dot{\theta} = \dot{\theta}^{\mathrm{el}} + \dot{\theta}^{\mathrm{vp}} = -\frac{1}{K}\frac{Dp}{Dt} + \dot{\lambda}\,\mathrm{tr}\left(\frac{\partial Q}{\partial \sigma_{ij}}\right). \tag{9}$$

Here the total volumetric strain rate is additively decomposed into an elastic and viscoplastic component, $K$ is the elastic bulk modulus, and $D/Dt$ is the material time derivative. A simple uni-axial idealization of both deviatoric and volumetric constitutive equations is illustrated in Fig. 1.



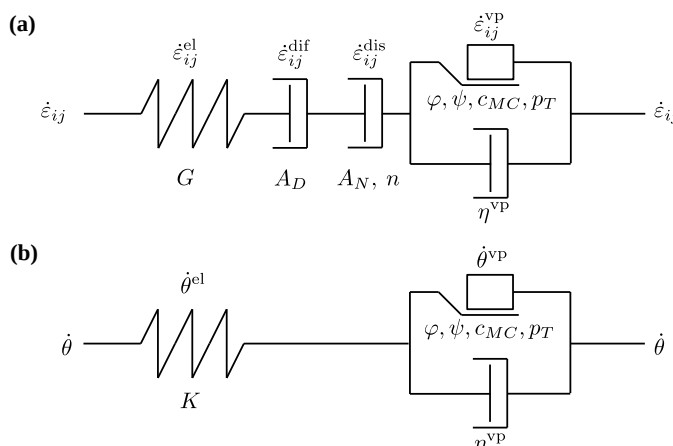

**Figure 1.** Schematic uni-axial representation of the rheological model. Elastic, viscous and plastic deformations are idealized by the spring, dashpot, and sliding frictional elements, receptively. **(a)** Deviatoric constitutive equation. **(b)** Volumetric constitutive equation.

In case the volumetric response becomes much stiffer compared to the deviatoric response (quasi-incompressible limit), it becomes mandatory to revert to a mixed formulation to compute accurate pressure fields. The continuity equation is discretized at the global level, and pressure is treated as a global independent variable. For incompressible Stokes flow, the continuity equation simply reduces to a constraint of the form $\dot{\theta} = 0$, and pressure becomes a corresponding Lagrange multiplier variable. Finally, the computed deviatoric stresses and pressure should satisfy the global equilibrium equation also known as Cauchy momentum equation, given by:

$$\frac{\partial \tau_{ij}}{\partial x_j} - \frac{\partial p}{\partial x_i} + \rho g_i = 0, \tag{10}$$

where $\rho$ is the material density, and $g_i$ are the components of the gravity acceleration vector. Here, we only consider quasi-static processes and hence explicitly ignore the inertial terms in the momentum equation.

## 2.3 Perzyna viscoplasticity

To deal with mesh dependency issues and convergence problems, which are typical for rate-independent plasticity models (e.g., Spiegelman et al., 2016; Duretz et al., 2019) we adopt a viscoplastic formulation following Perzyna (1966). In contrast with the Duvaut-Lions or consistency models, it has an explicit equation that defines the components of the viscoplastic strain rate. Here we use the simplest form of the Perzyna model without the power-law function, as it is defined in e.g. Jacquey and Cacace (2020a, b). The magnitude of the viscoplastic multiplier can be written as:

$$\dot{\lambda} = \frac{\langle F \rangle}{\eta^{\mathrm{vp}}}, \tag{11}$$





where $\eta^{\mathrm{vp}}$ is the viscoplastic regularization viscosity, $F$ is the rate-independent plastic yield function surrounded by Macaulay brackets, which has the following meaning:

$$\langle F \rangle = \begin{cases} F, & F \geq 0 \\ 0, & \text{otherwise} \end{cases}.$$ (12)

The rate-independent plasticity is naturally recovered with this formulation for $\eta^{\mathrm{vp}} \to 0$. Likewise, the viscoplastic deformation can be effectively switched off and the visco-elastic solution can be enforced with the limit $\eta^{\mathrm{vp}} \to \infty$.

Irrespective of the particular viscoplastic model, an addition of a regularization term implicitly introduces a length scale in strain localization problems (e.g. Duretz et al., 2023). It is important to note that regularization viscosity should not be treated as a part of the rheological model, but rather as a numerical parameter, which requires careful selection depending on the process being modeled (Duretz et al., 2019). There is also a trade-off between the sharpness of localization and number of nonlinear iterations required to achieve global convergence. In other words, increasing the amount of regularization improves the convergence but simultaneously smears out the localization zone. In the context of the Perzyna model, another issue requires additional attention. Direct inspection of Eq. (11) reveals that viscoplastic strain initiates only if stress violates the yield function constraint, i.e. a certain amount of *overstress* occurs. This feature imposes an explicit restriction on the yield (overstress) function to be continuous and convex in the entire stress space (Simo, 1989).

## 2.4 Smooth yield surface

Here we propose a simple two-surface plasticity model that combines a linear Drucker-Prager shear failure envelop with a circular tensile cap function in a manner that ensures its applicability in the context of Perzyna viscoplasticity (Fig. 2b). A similar model was proposed by Swan and Seo (2000), but continuity was only enforced at the yield surface, and dimensional consistency was violated due to usage of the squared versions of the yield functions. Our model requires four independent material parameters, which include the friction angle ($\varphi$), the dilation angle ($\psi$), the Mohr-Coulomb cohesion ($c_{\mathrm{MC}}$), and the tensile strength ($p_{\mathrm{T}}$). The other model parameters can be computed from the primary ones via geometrical transformations, which are omitted here for clarity. The overall parameter layout is schematically illustrated in Fig. 2d.

The Drucker-Prager cohesion ($c$), friction coefficient ($k$) and dilatation coefficient ($k_q$) are computed as follows:

$$k = \sin\varphi, \quad k_{\mathrm{q}} = \sin\psi, \quad c = c_{\mathrm{MC}} \cos\varphi.$$ (13)

This approach is rather simple, widely used in geodynamics (Keller et al., 2013; Kaus et al., 2016; Kaus, 2010; Duretz et al., 2021), and ensures that frictional strength is not systematically overestimated. For the details and alternative methods we refer to e.g. Jiang and Xie (2011). To ensure that the composite yield surface produces a continuous and differential map two issues need to be addressed: (i) a delimiter point between the segments must be selected such that they intersect in a smooth manner (Fig. 2a), and (ii) the tensile cap function must be scaled to eliminate the discontinuity outside the yield surface (Fig. 2b). We therefore introduce the following scaling coefficients, one for the yield surface ($a$), and one for the flow potential ($b$):

$$a = \sqrt{1 + k^2}, \quad b = \sqrt{1 + k_{\mathrm{q}}^2}.$$ (14)







**Figure 2.** Meridional plot of the smooth yield function and flow potential. **(a)** Yield function map without scaling ($a = 1$ in Eq.18). **(b)** Yield function map with scaling ($a \neq 1$ in Eq.18). **(c)** Flow potential map above the yield surface ($F > 0$). **(d)** Schematic illustration of the yield surface and flow potential parameters.

The center coordinate and the radius of the cap function, respectively, can be computed by enforcing the smooth intersection between both segments as:

$$
\quad p_{\mathrm{y}} = \left( p_{\mathrm{T}} + \frac{c}{a} \right) \left( 1 - \frac{k}{a} \right)^{-1}, \quad R_{\mathrm{y}} = p_{\mathrm{y}} - p_{\mathrm{T}}. \tag{15}
$$





The coordinates of the delimiter point are therefore given by:

$$p_\mathrm{d} = p_\mathrm{y} - R_\mathrm{y}\, \frac{k}{a}, \quad \tau_\mathrm{d} = k\, p_\mathrm{d} + c. \tag{16}$$

Finally, the center coordinate of the cap flow potential can be assigned such that the transition between the tensile and shear domains in the stress space passes through the delimiter point (see Fig. 2c), which yields:

$$p_\mathrm{q} = p_\mathrm{d} + k_\mathrm{q} \tau_\mathrm{d}. \tag{17}$$

Having defined all necessary model parameters we are ready to write down the equations for the yield function:

$$F = \begin{cases} \tau_\mathrm{II} - k\, p - c, & \tau_\mathrm{II}\, (p_\mathrm{y} - p_\mathrm{d}) \geq \tau_\mathrm{d}\, (p_\mathrm{y} - p) \\[2mm] a\left(\hat{R}_\mathrm{y} - R_\mathrm{y}\right), & \text{otherwise} \end{cases}, \tag{18}$$

and the corresponding flow potential:

$$Q = \begin{cases} \tau_\mathrm{II} - k_\mathrm{q}\, p - \mathrm{const}, & \tau_\mathrm{II}\, (p_\mathrm{q} - p_\mathrm{d}) \geq \tau_\mathrm{d}\, (p_\mathrm{q} - p) \\[2mm] b\left(\hat{R}_\mathrm{q} - \mathrm{const}\right), & \text{otherwise} \end{cases}. \tag{19}$$

Here the radii of the points in the stress space for both the tensile yield function and flow potential, respectively, are given by:

$$\hat{R}_\mathrm{y} = \sqrt{\tau_\mathrm{II}^2 + (p - p_\mathrm{y})^2}, \quad \hat{R}_\mathrm{q} = \sqrt{\tau_\mathrm{II}^2 + (p - p_\mathrm{q})^2}. \tag{20}$$

It should be pointed out that our model gives a simple analytical description of the stress space partitioning into shear and tensile domains, and therefore no complex algorithm is required to detect the active yield surface. In any point above the yield surface the gradient of the flow potential is uniquely defined, and hence the return mapping direction is completely constrained.
This property greatly simplifies the implementation of the stress update algorithms. Since an arbitrary amount of dilatation is allowed in our model, the resulting flow potential function is non-associated, as is illustrated in Fig. 2c. The dilatation angle for the Drucker-Prager part of the yield function does not affect the description of the tensile failure, which implies that zero dilatation angle cases are explicitly supported.

We complete the formulation of the plasticity model by directly differentiating Eq. (19) to obtain the required expressions
for the flow potential gradients:

$$\frac{\partial Q}{\partial \sigma_{ij}} = B_\tau \tau_{ij} + B_p \delta_{ij}, \tag{21}$$

where the prefactors are given by:

$$(B_\tau, B_p) = \begin{cases} \left(\frac{1}{2\tau_\mathrm{II}}, \frac{k_\mathrm{q}}{3}\right), & \tau_\mathrm{II}\, (p_\mathrm{q} - p_\mathrm{d}) \geq \tau_\mathrm{d}\, (p_\mathrm{q} - p) \\[3mm] \left(\frac{b}{2\hat{R}_\mathrm{q}}, -\frac{b(p - p_\mathrm{q})}{3\hat{R}_\mathrm{q}}\right), & \text{otherwise} \end{cases}. \tag{22}$$





## 2.5 Strain softening

Plastic deformation in the rocks and soils is generally characterized by the occurrence of the strain softening (see the extensive review by Read and Hegemier, 1984, for the physical details and explanations of this phenomenon) . Here, we parametrize strain softening for the Mohr-Coulomb friction angle and cohesion as a linear function of the accumulated deviatoric viscoplastic strain, defined as:

$$\kappa = \int_t \dot{\varepsilon}_{\mathrm{II}}^{\mathrm{vp}} \, \mathrm{dt}. \tag{23}$$

The expressions for the friction angle and cohesion can be formulated as follows:

$$\varphi = \max\left(\varphi^{\mathrm{init}} + H_\varphi \kappa, \, \varphi^{\mathrm{min}}\right), \quad c_{\mathrm{MC}} = \max\left(c_{\mathrm{MC}}^{\mathrm{init}} + H_c \kappa, \, c_{\mathrm{MC}}^{\mathrm{min}}\right), \tag{24}$$

where the superscripts $^{\mathrm{init}}$ and $^{\mathrm{min}}$ denote the initial and saturated values for the corresponding material parameters, $H_\varphi$ and $H_c$ are the hardening modulii for the friction angle and cohesion, respectively. Note that the hardening modulii are negative for softening cases.

To identify zones of tensile failure and dilatant shear plasticity the accumulated volumetric viscoplastic strain can be defined in a similar way:

$$\chi = \int_t \dot{\theta}^{\mathrm{vp}} \, \mathrm{dt}. \tag{25}$$

## 3 Numerical formulation

### 3.1 Time discretization

To integrate the coupled nonlinear constitutive equations in time we apply backward Euler implicit time discretization which is unconditionally stable and first order accurate. All rate quantities are presented by their unknown values at the end of the time step and assumed to be constant during the time step. In that case the computation of the incremental quantities can be done trivially, e.g. for the deviatoric strain rate tensor we can write:

$$\Delta\varepsilon_{ij} = \dot{\varepsilon}_{ij} \, \Delta t, \tag{26}$$

where $\Delta t = t_{n+1} - t_n$ is the time step, $t_n$ is the time in the beginning, and $t_{n+1}$ is the time in the end of the $n$-th time step. Similarly, the rate quantities can be estimated from their corresponding increments as:

$$\dot{\varepsilon}_{ij} = \frac{\Delta\varepsilon_{ij}}{\Delta t}. \tag{27}$$

To simplify the notation we omit the indices at the end of the time step for all quantities in the remainder of this paper, e.g. for the pressure we write $p$ instead of $p_{n+1}$. For the converged values from the previous time step we always include the time step

index, e.g. for the history pressure we write $p_n$.





## 3.2 True pressure scheme

In the context of a mixed formulation, the globally discretized velocity (displacement increment) and pressure are typically treated as primary unknowns. Since dilatant plasticity, in general, involves pressure modifications during local stress updates, we must decide how to interpret the global pressure variable (which is obtained from solving the global system of nonlinear

equations). One of the approaches is based on treating the global pressure as a true spherical part of the Cauchy stress tensor (Commend et al., 2004). Hence, the deviatoric stress, local pressure, and viscoplastic volumetric strain rate must be computed using a standard strain-driven approach, solely as the functions of the strain rate, and hence the velocity:

$$\tau_{ij} = \tau_{ij}\left(\dot{\epsilon}_{ij}\right), \quad p = p\left(\dot{\epsilon}_{ij}\right), \quad \dot{\theta}^{\mathrm{vp}} = \dot{\theta}^{\mathrm{vp}}\left(\dot{\epsilon}_{ij}\right). \tag{28}$$

At this stage, the globally discretized pressure, denoted as $p^*$, may differ from the locally computed pressure in the integration

points, i.e.: $p^* \neq p$. Since the global pressure is treated as a true pressure variable, the local pressure should be discarded, and only the deviatoric stress and viscoplastic volumetric strain rate should be used for the evaluation of the time-discrete momentum and continuity residual equations, respectively:

$$r_i^{\mathrm{m}} = \frac{\partial \tau_{ij}}{\partial x_j} - \frac{\partial p^*}{\partial x_i} + \rho g_i,$$
$$r^{\mathrm{c}} = \dot{\theta} + \frac{1}{K}\frac{p^* - p_n}{\Delta t} - \dot{\theta}^{\mathrm{vp}}. \tag{29}$$

The addition of the viscoplastic strain rate to the continuity residual ensures that the global pressure receives a feedback from

the volumetric viscoplasticity. Upon convergence of the global nonlinear iterations, both pressure variables should become approximately equal, i.e.: $p^* \approx p$.

## 3.3 Trial pressure scheme

The globally discretized pressure variable $(p^*)$ can alternatively be interpreted as a trial visco-elastic pressure (Duretz et al., 2021). In this case, the deviatoric stress, local pressure, and viscoplastic volumetric strain rate become functions of both strain

rate and global pressure:

$$\tau_{ij} = \tau_{ij}\left(\dot{\epsilon}_{ij}, p^*\right), \quad p = p\left(\dot{\epsilon}_{ij}, p^*\right), \quad \dot{\theta}^{\mathrm{vp}} = \dot{\theta}^{\mathrm{vp}}\left(\dot{\epsilon}_{ij}, p^*\right). \tag{30}$$

Specifically, the pressure update assumes the following discrete form in this formulation (Duretz et al., 2021):

$$p = p^* + K\,\dot{\theta}^{\mathrm{vp}}\,\Delta t. \tag{31}$$

We first formulate the time-discrete residual equations involving both the updated deviatoric stress and pressure:

$$r_i^{\mathrm{m}} = \frac{\partial \tau_{ij}}{\partial x_j} - \frac{\partial p}{\partial x_i} + \rho g_i, \tag{32}$$

$$r^{\mathrm{c}} = \dot{\theta} + \frac{1}{K}\frac{p - p_n}{\Delta t} - \dot{\theta}^{\mathrm{vp}}. \tag{33}$$



Substituting Eq. (31) into the continuity residual equation (Eq. 33) yields:

$$r^{\mathrm{c}} = \dot{\theta} + \frac{1}{K}\frac{p^* - p_n}{\Delta t}. \tag{34}$$

The viscoplastic volumetric strain rate simply cancels from the global continuity residual. The reason for this is that the local
pressure update (Eq. 31) is derived by enforcing the continuity equation in its complete form (Eq. 33). This implies that the
continuity residual (Eq. 33) is always balanced at the level of the local stress update. The local pressure ($p$) represents the true
spherical part of the Cauchy stress tensor upon achieving global convergence. The difference between the global and local
pressure does not vanish, i.e.: $p^* \neq p$. The global pressure ($p^*$) converges to the trial pressure value. The global solver should
therefore directly apply spatial discretization to the Eqs. (32) and (34). Using the Eq. (33) in the global solver is not justified,
since it is indistinguishable from the Eq. (34).

### 3.4 Pressure scheme comparison

Both pressure schemes should theoretically deliver the same results. However their convergence properties are not guaranteed
to be the same. First, it should be noted that for non-dilatant plasticity cases, the difference between the pressure variables does
not occur in either of the approaches, i.e. $p^* = p$. It might also be taken for granted that both schemes should deliver the same
linearization in this case. This assumption does not hold when the yield function still depends on pressure, i.e. $F = F(p)$, such
as for Drucker-Prager plasticity with a zero dilatation coefficient. The true pressure scheme would still imply that pressure
used to evaluate the yield function is estimated from the strain rates, i.e.: $p = p(\dot{\epsilon}_{ij})$. The trial pressure scheme would instead
directly use the global pressure ($p^*$). Even when both pressure values should theoretically be the same, at least for stable
discretizations, this still formally leads to different functional dependencies, different linearizations, and different convergence
properties. Only for the truly pressure-independent plasticity models, like the Von Mises model, both schemes are guaranteed to
be exactly same (which is irrelevant in the context of rock failure modeling). It should be also noted that both pressure schemes
always deliver non-symmetric global tangent matrices for the pressure-dependent plasticity models. This statement holds even
for completely associated models, when the friction angle is equal to the dilatation angle. In general, all dilatant plasticity
models require the elastic bulk modulus ($K$) to be finite, and both pressure schemes will fail to handle dilatant plasticity cases
for elastically incompressible materials ($K \to \infty$).

A detailed analysis of the convergence properties of both pressure schemes goes beyond the scope of this paper. Here, we
simply report our practical observations and conclusions. We have thoroughly implemented, linearized and tested both pressure
schemes for the relevant dilatant and non-dilatant plasticity cases (see Berlie, 2023, for further details). The results demonstrate
that the trial pressure scheme performs more robustly compared to the true pressure formulation. In the following, we therefore
limit our discussion to the trial pressure scheme.

### 3.5 Primary variable selection

Traditionally, velocity is selected as the primary kinematical variable to solve boundary value problems that involve quasi-
static deformation of viscous materials. However, this is a suboptimal choice in the presence of highly nonlinear rheologies



and Dirichlet boundary conditions. The problem is caused by the lack of time step control over the magnitude of the kine-
matical loading. With a velocity formulation, the entire boundary velocity is applied instantly, which may result in severe
convergence problems. There is essentially no measure that can be applied to mitigate this problem, since boundary velocities
are independent of the time step. We therefore choose to use displacement increments ($\Delta u_i$) rather than velocity as the primary
kinematical variables in this work. This approach has the advantage that kinematical loading can be directly controlled by the
time step, since the velocity and displacement increments are related in the following manner:

$$\Delta u_i = v_i \, \Delta t. \tag{35}$$

This property opens the possibility to design an adaptive time stepping algorithm. As soon as the selected time step is too
large to achieve convergence within a reasonable number of iterations, the equilibrium iteration can be simply restarted with a
smaller time step magnitude, which implies a smaller displacement step and therefore a smaller stress increment.

We emphasize that using displacement increments instead of velocities imposes no restrictions on the type of rheology that
can be handled (viscous, elastic, viscoplastic, or combinations thereof). Both velocities and strain rates are perfectly defined
in this formulation (see Eq. 27). Hence the stress integration algorithm can be formulated using strain rates. For instance the
algorithm presented in Sect. 3.6 can be readily used in the framework of a finite element or a finite difference code that uses a
standard approach with the velocity as a primary variable.

As the second global primary variable in the mixed formulation, we select the trial pressure increment ($\Delta p^*$). The total trial
pressure is thus defined as:

$$p^* = p_n + \Delta p^*. \tag{36}$$

### 3.6  Local stress update algorithm

The integration of constitutive equations that include a viscoplastic component is usually done with a two-stage predictor-
corrector procedure (see Appendix A for derivations). During the first stage, the magnitude of the viscoplastic multiplier ($\dot{\lambda}$) is
assumed to be zero and the entire strain rate is visco-elastic (*predictor stage*). The magnitude of the trial visco-elastic deviatoric
stress ($\tau_{\mathrm{II}}^{\mathrm{ve}}$) can be obtained by solving the following scalar nonlinear residual equation using the Newton-Raphson method (see
Eq. A7):

$$r^{\mathrm{ve}} = \dot{\varepsilon}_{\mathrm{II}}^* - A_{\mathrm{L}} \, \tau_{\mathrm{II}}^{\mathrm{ve}} - A_{\mathrm{N}} \left(\tau_{\mathrm{II}}^{\mathrm{ve}}\right)^n = 0, \tag{37}$$

where the $\dot{\varepsilon}_{\mathrm{II}}^*$ is the effective deviatoric strain rate that depends on the elastic stresses from the previous time step

$$\dot{\varepsilon}_{ij}^* = \dot{\varepsilon}_{ij} + \frac{\tau_{ij}^*}{2G\Delta t}, \tag{38}$$

and $A_{\mathrm{L}}$ is the effective linear stress term prefactor, which combines elastic shear and diffusion creep constants:

$$A_{\mathrm{L}} = \frac{1}{2G\Delta t} + A_{\mathrm{D}}. \tag{39}$$





Here $\tau_{ij}^*$ denotes the rotated deviatoric stress from the previous time step (see the Appendix A for more details). Note that stress

rotation terms are usually computed using an alternative incrementally-objective scheme (Thielmann et al., 2015; Gerya, 2019),

which produces asymptotically correct results for finite time steps. The 3D generalization of this algorithm was originally

provided by Rubinstein and Atluri (1983). The detailed explanation why using the Jaumann objective stress rate does not lead

to the spurious stress oscillation for the rocks is provided in Thielmann et al. (2015). The derivative of the visco-elastic scalar

residual with respect to the magnitude of the visco-elastic deviatoric stress is readily evaluated as:

$$\frac{\partial r^{\text{ve}}}{\partial \tau_{\text{II}}^{\text{ve}}} = -A_{\text{L}} - n A_{\text{N}} (\tau_{\text{II}}^{\text{ve}})^{n-1}. \tag{40}$$

To ensure robust convergence of the nonlinear iteration we use the following initial guess:

$$\tau_{\text{II}}^{\text{ve}} \approx \left(\frac{1}{\tau_L} + \frac{1}{\tau_N}\right)^{-1}, \quad \tau_L = \frac{\dot{\varepsilon}_{\text{II}}^*}{A_L}, \quad \tau_N = \left(\frac{\dot{\varepsilon}_{\text{II}}^*}{A_N}\right)^{1/n}, \tag{41}$$

which represents the quasi-harmonic average of two closed-form solutions obtained for each isolated creep mechanism.

During the second stage, the yield function is evaluated using the trial stresses, i.e.: $F = F(\tau_{\text{II}}^{\text{ve}}, p^*)$. If the trial yield function

is not violated ($F < 0$), the visco-elastic solution is accepted, i.e.: $\tau_{II} = \tau_{II}^{\text{ve}}$, otherwise the viscoplasticity stress constraints

are enforced (*corrector stage*). This step requires solving a coupled system of nonlinear equations with the residual vector (**r**)

incorporating deviatoric (Eq. A7), continuity (Eq. A14) and viscoplastic constraint residuals (Eq. A17). The corresponding

solution vector (**x**) contains the effective deviatoric stress ($\tau_{\text{II}}$), the local pressure ($p$) and the viscoplastic multiplier ($\dot{\lambda}$), as

primary unknowns:

$$\mathbf{r} = \begin{bmatrix} \dot{\varepsilon}_{\text{II}}^* - A_{\text{L}} \tau_{\text{II}} - A_{\text{N}} (\tau_{\text{II}})^n - \dot{\lambda} A_\tau \\ \frac{p - p^*}{K \Delta t} - \dot{\lambda} A_p \\ \langle F \rangle - \dot{\lambda} \eta^{\text{vp}} \end{bmatrix} = \begin{bmatrix} 0 \\ 0 \\ 0 \end{bmatrix}, \quad \mathbf{x} = \begin{bmatrix} \tau_{\text{II}} \\ p \\ \dot{\lambda} \end{bmatrix}. \tag{42}$$

Here the deviatoric and volumetric plastic constants, respectively, are formulated as follows:

$$(A_\tau, A_p) = \begin{cases} \left(\frac{1}{2}, k_{\text{q}}\right), & \tau_{\text{II}} (p_{\text{q}} - p_{\text{d}}) \geq \tau_{\text{d}} (p_{\text{q}} - p) \\ \left(\frac{b \tau_{\text{II}}}{2 \hat{R}_{\text{q}}}, -\frac{b(p - p_{\text{q}})}{\hat{R}_{\text{q}}}\right), & \text{otherwise} \end{cases}. \tag{43}$$

A Newton-Raphson scheme is employed to update the solution until a sufficiently small tolerance is obtained:

$$\mathbf{x}_{k+1} = \mathbf{x}_k - \alpha \mathbf{J}_k^{-1} \mathbf{r}_k, \quad \|\mathbf{r}_k\| < \text{tol}, \tag{44}$$

where $k$ is the iteration index, **J** the local Jacobian matrix of the local stress update, and $\alpha_{\min} < \alpha < 1$ the step length that is

optimally selected by a simple back-tracking line-search algorithm guided by the Armijo rule (Armijo, 1966) to ensure that the

nonlinear residual is sufficiently reduced between the iterations, i.e:

$$\|\mathbf{r}_{k+1}\| \leq \gamma \|\mathbf{r}_k\|, \tag{45}$$





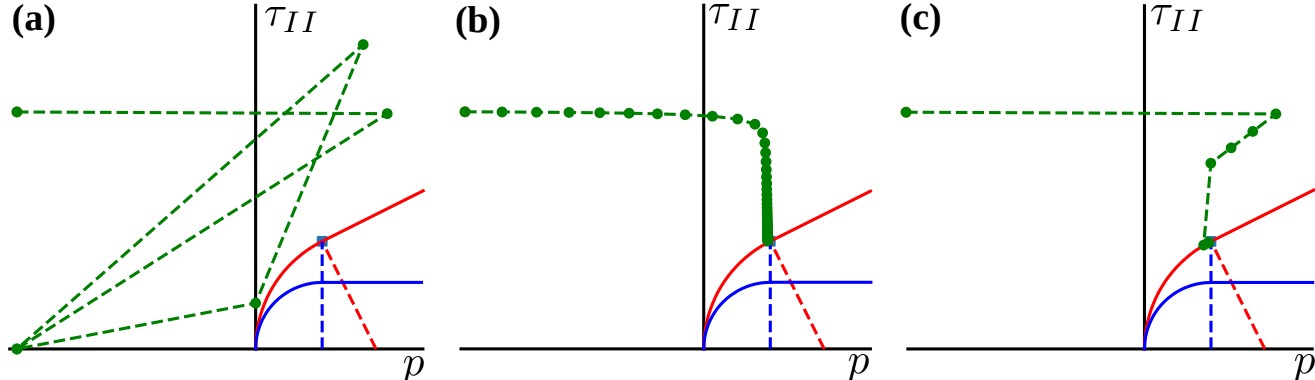

**Figure 3.** Convergence pattern of the nonlinear local iterations for the corrector stage ($\eta^{\mathrm{vp}} = 0$) when the initial guess is relatively far away from the yield surface. **(a)** Full Newton steps are applied ($\alpha = 1$). Closed loops and divergence of the algorithm is observed. **(b)** Damped Newton steps are applied throughout ($\alpha = 0.1$), which results in a smooth pattern, but in a relatively large number of iterations. **(c)** The step length is controlled by a line search algorithm ($\gamma = 0.9$, $\alpha_{\min} = 0.1$). Fast convergence is obtained.

where $0 < \gamma < 1$ is the residual reduction parameter, which is typically assigned to a rather large value $\gamma = 0.9$. To ensure progress of the nonlinear iteration, the step length is bounded by a lower threshold $\alpha_{\min}$. We explicitly emphasize the impor-
tance of the line search for the stability of the local iteration algorithms. Despite having a completely smooth yield surface and flow potential, the local iterations can still produce closed loops in the stress space that never lead to convergence when full steps are applied (Fig.3a). A line search algorithm resolves this issue (Fig.3c).

To initialize the nonlinear iteration we use the trial visco-elastic stresses and assign the viscoplastic multiplier to zero:

$$\mathbf{x}_0 = \begin{bmatrix} \tau_{\mathrm{II}}^{\mathrm{ve}} \\ p^* \\ 0 \end{bmatrix}. \tag{46}$$

Finally, the Jacobian matrix can be obtained by differentiating the residuals with respect to the unknowns:

$$\mathbf{J} = \begin{bmatrix} -A_{\mathrm{L}} - n\,A_{\mathrm{N}}\,(\tau_{\mathrm{II}})^{n-1} - \dot{\lambda}\,\frac{\partial A_\tau}{\partial \tau_{\mathrm{II}}} & -\dot{\lambda}\,\frac{\partial A_\tau}{\partial p} & -A_\tau \\ -\dot{\lambda}\,\frac{\partial A_p}{\partial \tau_{\mathrm{II}}} & \frac{1}{K\Delta t} - \dot{\lambda}\,\frac{\partial A_p}{\partial p} & -A_p \\ \frac{\partial F}{\partial \tau_{\mathrm{II}}} & \frac{\partial F}{\partial p} & -\eta^{\mathrm{vp}} \end{bmatrix}, \tag{47}$$

where the required stress derivatives of the plastic constants ($A_\tau$), ($A_p$) and the yield function ($F$) are given by Eqs. (A9), (A16), and (A18), respectively. After convergence of the local iterations, the deviatoric stresses are obtained as follows:

$$\tau_{ij} = \tau_{\mathrm{II}}\,n_{ij}, \tag{48}$$





where the normalized deviatoric direction tensor is defined as:

$$n_{ij} = \frac{\dot{\varepsilon}^*_{ij}}{\dot{\varepsilon}^*_{\mathrm{II}}}. \tag{49}$$

The corresponding update of the accumulated deviatoric viscoplastic strain is given by:

$$\kappa = \kappa_n + \Delta\kappa, \quad \Delta\kappa = \dot{\lambda}\, A_\tau \Delta t. \tag{50}$$

To simplify the presentation of the material model in this paper we apply the strain softening explicitly between the time steps.
During the local stress update, the yield function is evaluated using the $\kappa$ values in the beginning of the time step, i.e. $F(\kappa_n)$.
The plastic strength parameters are instantly updated according the Eqs. (50) and (24) once the converged values of $\Delta\kappa$ are
computed. However, the presented algorithm can be easily modified to include the viscoplastic strain evolution in the local
stress update in a coupled manner. This would imply updating $\Delta\kappa$ during the iterations and using $F(\kappa)$ instead. Finally, the
accumulated volumetric viscoplastic strain is updated as follows:

$$\chi = \chi_n + \Delta\chi, \quad \Delta\chi = \dot{\lambda}\, A_p \Delta t. \tag{51}$$

### 3.7 Finite element formulation

We adopt the conforming Crouzeix-Raviart triangular finite element (Crouzeix and Raviart, 1973) to discretize the momentum
and continuity residual equations (Eqs. 32 and 34) in the framework of a standard Galerkin procedure. The element is LBB-
stable and behaves robustly in practice. Since significant accuracy deterioration was previously reported for the isoparametric
finite elements with curvilinear edges (e.g. Lee and Bathe, 1993), we adopt a subparametric formulation and keep the edges of
the elements straight. For the accurate evaluation of the element integrals we use an efficient Gaussian quadrature rule specif-
ically designed for triangular elements (Dunavant, 1985). The 2D element has 6 integration points, where all the constitutive
equations are evaluated and all the history variables are stored. The details of the finite element discretization are illustrated in
Fig. 4.

In the reminder of this section we summarize the details of the finite element discretization using the standard matrix notation
(Zienkiewicz and Taylor, 2000). The primary variables are interpolated in the integration points as follows:

$$\boldsymbol{\Delta u} = \mathbf{N_u}\, \boldsymbol{\Delta u_e}, \quad \Delta p^* = \mathbf{N_p}\, \boldsymbol{\Delta p^*_e}, \tag{52}$$

where $\boldsymbol{\Delta u_e}$ and $\boldsymbol{\Delta p^*_e}$ are the displacement and the trial pressure increment vectors of the element, respectively, and the
corresponding interpolation matrices are given by:

$$\mathbf{N_u} = \begin{bmatrix} N_{u1} & 0 & \ldots & N_{u7} & 0 \\ 0 & N_{u1} & \ldots & 0 & N_{u7} \end{bmatrix}, \quad \mathbf{N_p} = \begin{bmatrix} N_{p1} & \ldots & N_{p3} \end{bmatrix}. \tag{53}$$

Here $N_{ui}$ and $N_{pi}$ are respectively the displacement shape function and the pressure shape function of the i-th node.




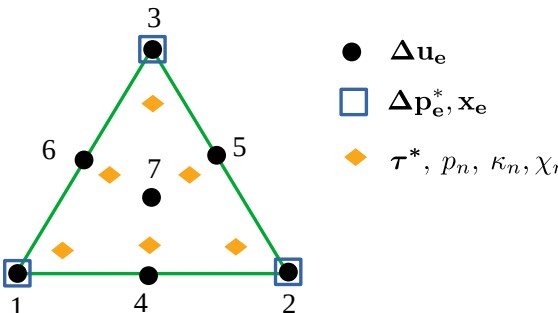

**Figure 4.** Conforming triangular Crouzeix-Raviart finite element ($P_2^+ \times P_{-1}$). Displacement increments ($\mathbf{\Delta u_e}$) are interpolated using quadratic shape functions enhanced with a bubble function in the central node. Pressure increments ($\mathbf{\Delta p_e^*}$) are interpolated using linear shape functions and assumed to be discontinuous between elements. Element coordinates ($\mathbf{x_e}$) are stored only in the element corners, and edges remain straight (subparametric element). History variables and material parameters are stored in the integration points.

We use Voigt notation to represent the second order symmetric tensors in the vector form. Hence for the total strain increment ($\mathbf{\Delta\epsilon}$), the deviatoric strain increment ($\mathbf{\Delta\varepsilon}$), and the deviatoric stress ($\boldsymbol{\tau}$) we can write:

$$\mathbf{\Delta\epsilon} = \begin{bmatrix} \Delta\epsilon_{xx} \\ \Delta\epsilon_{yy} \\ \Delta\epsilon_{zz} \\ \Delta\gamma_{xy} \end{bmatrix}, \quad \mathbf{\Delta\varepsilon} = \begin{bmatrix} \Delta\varepsilon_{xx} \\ \Delta\varepsilon_{yy} \\ \Delta\varepsilon_{zz} \\ \Delta\varepsilon_{xy} \end{bmatrix}, \quad \boldsymbol{\tau} = \begin{bmatrix} \tau_{xx} \\ \tau_{yy} \\ \tau_{zz} \\ \tau_{xy} \end{bmatrix}, \tag{54}$$

where $\gamma_{xy}$ is the engineering shear strain, which is twice larger than the corresponding tensor component, i.e. $\gamma_{xy} = 2\,\epsilon_{xy}$. The total strain increment vector is computed as:

$$\mathbf{\Delta\epsilon} = \mathbf{B}\,\mathbf{\Delta u_e}, \quad \mathbf{B} = \begin{bmatrix} \frac{\partial N_{u1}}{\partial x} & 0 & \dots & \frac{\partial N_{u7}}{\partial x} & 0 \\ 0 & \frac{\partial N_{u1}}{\partial y} & \dots & 0 & \frac{\partial N_{u7}}{\partial y} \\ 0 & 0 & \dots & 0 & 0 \\ \frac{\partial N_{u1}}{\partial y} & \frac{\partial N_{u1}}{\partial x} & \dots & \frac{\partial N_{u7}}{\partial y} & \frac{\partial N_{u7}}{\partial x} \end{bmatrix}. \tag{55}$$

Here $\mathbf{B}$ is the differential operator matrix that contains the derivatives of the displacement shape functions with respect to the global coordinates. Note that the third row explicitly enforces the plane strain kinematical constraint $\epsilon_{zz} = 0$. The deviatoric and volumetric strain increments, respectively, are given by:

$$\mathbf{\Delta\varepsilon} = \mathbf{I}^D\,\mathbf{\Delta\epsilon}, \quad \Delta\theta = \mathbf{m}^T\,\mathbf{\Delta\epsilon}, \tag{56}$$





where the corresponding projection matrices are as follows:

$$\mathbf{I}^D = \mathbf{I} - \frac{1}{3}\mathbf{m}\,\mathbf{m}^T, \quad \mathbf{I} = \frac{1}{2}\begin{bmatrix} 2 & & & \\ & 2 & & \\ & & 2 & \\ & & & 1 \end{bmatrix}, \quad \mathbf{m} = \begin{bmatrix} 1 \\ 1 \\ 1 \\ 0 \end{bmatrix}. \tag{57}$$

The history variables from the previous time step are assumed to be stored directly in the integration points illustrated as
diamonds in Fig. 4. These include the rotated history stress ($\boldsymbol{\tau}^*$), the converged true pressure ($p_n$), and the accumulated deviatoric ($\kappa_n$) and volumetric ($\chi_n$) viscoplastic strains. The following estimate for the effective deviatoric strain rate is readily available:

$$\dot{\boldsymbol{\varepsilon}}^* = \frac{\boldsymbol{\Delta\varepsilon}}{\Delta t} + \frac{\boldsymbol{\tau}^*}{2G\Delta t}. \tag{58}$$

Combining the Eqs. (36) and (52) we can express the total trial pressure in the integration point as:

$$p^* = p_n + \mathbf{N_p}\,\boldsymbol{\Delta p_e^*}. \tag{59}$$

Finally, the scalar norm of the effective deviatoric strain rate ($\dot{\varepsilon}_{\mathrm{II}}^*$), and the unit deviatoric direction ($\mathbf{n}$) can be evaluated as:

$$\dot{\varepsilon}_{\mathrm{II}}^* = \sqrt{\frac{1}{2}\left(\dot{\varepsilon}_{xx}^{*2} + \dot{\varepsilon}_{yy}^{*2} + \dot{\varepsilon}_{zz}^{*2}\right) + \dot{\varepsilon}_{xy}^{*2}}, \quad \mathbf{n} = \frac{\dot{\boldsymbol{\varepsilon}}^*}{\dot{\varepsilon}_{II}^*}. \tag{60}$$

At this stage, we are ready to invoke the local stress update algorithm summarized in Sect. 3.6 to compute the deviatoric stress ($\boldsymbol{\tau}$) and true pressure ($p$), and hence the total Cauchy stress:

$$\boldsymbol{\sigma} = \boldsymbol{\tau} - p\,\mathbf{m}. \tag{61}$$

The weak forms of the time-discrete momentum and continuity residual equations (Eqs. 32 and 34) are given by:

$$\mathbf{r_u} = \int\limits_V \mathbf{B}^T\boldsymbol{\sigma}\,dV - \int\limits_V \mathbf{N_u}^T\,\mathbf{b}\,dV - \int\limits_S \mathbf{N_u}^T\,\mathbf{t}\,dS = 0,$$

$$\tag{62}$$

$$\mathbf{r_p} = -\int\limits_V \mathbf{N_p}^T\,\Delta\theta\,dV - \int\limits_V \mathbf{N_p}^T\,\frac{\Delta p^*}{K}\,dV = 0,$$

where $\mathbf{b} = \rho\mathbf{g}$ are the gravity forces, and $\mathbf{t}$ is the traction vector, which is integrated over the surface. The standard Newton-Raphson iteration is employed to update the primary variables until the global convergence is achieved:

$$\begin{bmatrix} \boldsymbol{\Delta} u \\ \boldsymbol{\Delta p}^* \end{bmatrix}_{k+1} = \begin{bmatrix} \boldsymbol{\Delta} u \\ \boldsymbol{\Delta p}^* \end{bmatrix}_k - \begin{bmatrix} \mathbf{K_{uu}} & \mathbf{K_{up}} \\ \mathbf{K_{pu}} & \mathbf{K_{pp}} \end{bmatrix}_k^{-1} \begin{bmatrix} \mathbf{r_u} \\ \mathbf{r_p} \end{bmatrix}_k. \tag{63}$$





Here $k$ denotes the iteration index, and the blocks of the coupled tangent matrix are evaluated as follows:

$$\mathbf{K_{uu}} = \int_V \mathbf{B}^T \mathbf{D} \mathbf{B} \, dV, \qquad \mathbf{K_{up}} = -\int_V \mathbf{B}^T \mathbf{q} \mathbf{N_p} \, dV,$$

$$\mathbf{K_{pu}} = -\int_V \mathbf{N_p}^T \mathbf{m}^T \mathbf{B} \, dV, \quad \mathbf{K_{pp}} = -\int_V \mathbf{N_p}^T \frac{1}{K} \mathbf{N_p} \, dV, \tag{64}$$

where the tangent operators are given by:

$$\mathbf{D} = \frac{\partial \boldsymbol{\sigma}}{\partial \boldsymbol{\Delta \epsilon}} = \left(2\,\eta_{\text{eff}} \,\mathbf{I}^D + \beta_1 \,\mathbf{n}\,\mathbf{n}^T + \beta_2 \,\mathbf{m}\,\mathbf{n}^T\right) / \Delta t,$$

$$\mathbf{q} = \frac{\partial \boldsymbol{\sigma}}{\partial p^*} = \beta_3 \,\mathbf{n} + \beta_4 \mathbf{m}. \tag{65}$$

Details of the derivations along with expressions for the coefficients $\eta_{\text{eff}}$, and $\beta_1 - \beta_4$ are provided in Appendix B. It should be noted that only completely pressure-independent constitutive models (such as von Mises plasticity or nonlinear visco-elasticity) produce a symmetric tangent matrix. In these cases, the stiffness coefficients assume the following values: $\beta_2 = \beta_3 = 0$, $\beta_4 = 1$. Any form of pressure dependence in the constitutive model, including the fully-associated plasticity case, introduces non-symmetric terms in the tangent matrix. This property of the mixed two-field formulation contrasts with the classical strain-driven approach which preserves symmetry for the associated pressure-dependent plasticity models.

The global Newton iteration is terminated as soon as the following stopping criterion is satisfied:

$$\frac{\left\|\mathbf{f^{int}} - \mathbf{f^{ext}}\right\|}{\left\|\mathbf{f^{ext}}\right\|} < \text{tol}, \tag{66}$$

where the partition of the momentum residual vector into the external and internal forces is defined as:

$$\mathbf{f^{int}} = \int_V \mathbf{B}^T \boldsymbol{\sigma} \, dV, \quad \mathbf{f^{ext}} = \int_V \mathbf{N_u}^T \mathbf{b} \, dV + \int_S \mathbf{N_u}^T \mathbf{t} \, dS = 0 \tag{67}$$

There is no need to explicitly check the residual of the continuity equation, since it is typically satisfied to a very low tolerance at every global iteration.

After achieving global convergence, velocities, deviatoric strain rates, and volumetric strain rates, respectively, can be computed for postprocessing purposes using:

$$\mathbf{v} = \frac{\boldsymbol{\Delta} u}{\Delta t}, \quad \dot{\varepsilon} = \frac{\boldsymbol{\Delta} \varepsilon}{\Delta t}, \quad \dot{\theta} = \frac{\Delta \theta}{\Delta t}. \tag{68}$$

## 3.8 Software implementation

To test the applicability and robustness of the yield surface developed in this work, we implemented the algorithms presented in Sect. 3.6 and 3.7 in a compact Python finite element code (GeoTech2D) that employs LBB-stable triangular Crouziex-Raviart elements. We use an efficient array implementation from the NumPy package (Harris et al., 2020). The finite element matrices





are initially stored in coordinate format and subsequently assembled in a compressed sparse column format using the SciPy package (Virtanen et al., 2020). Dabrowski et al. (2008) implemented a similar approach and demonstrated that it has superior efficiency compared to a direct assembly. The default sparse direct solver provided by the SciPy package is used to solve the linearized system of equations. We do not utilize the block structure of the tangent matrix by adopting efficient approaches such as Powell-Hestenes iterations (e.g. Dabrowski et al., 2008). Since we only consider elastically compressible cases the robustness of the direct solver is sufficient to perform a coupled factorization. To improve the robustness and speed of the code, iterative solvers could be added at later stage.

To generate the unstructured triangular grids we use the Triangle mesh generator (Shewchuk, 1996), which we invoke via the Python interface implemented in the MeshPy package (Kloeckner et al., 2022). In the supplement we provide all the scripts that we used to generate the grids in this paper along with an example script that demonstrates the basic features of the Triangle interface in MeshPy, such as material regions, holes, and boundary markers. We use the pyEVTK package (Herrera, 2021) to store the simulation results in the Visualization Toolkit (VTK) format (Schroeder et al., 2006), which can be visualized with the ParaView package (Ahrens et al., 2005). Whenever necessary we use the scientific color maps (Crameri et al., 2020), which prevent visual distortion of the data.

## 4    Benchmarks and applications

In this section we test the application of the developed plasticity model to a number of relevant problems that involve both mode-I and mode-II plastic failure. The material parameters, temporal and spatial discretization details, domain geometry, and boundary conditions are described in Table 1. Every problem we consider here is summarized in a single column of this table, which is labeled with a keyword concisely describing the essence of the setup (e.g. *Crust* or *Tensile*, etc). References to the figures that demonstrate basic results of a corresponding numerical simulation are also provided. Parameters related to different scenarios of the considered problems (e.g. extension vs. compression) are separated by a slash.

### 4.1    0D stress integration test

The correctness of the stress integration algorithm implementation can be demonstrated by performing the so-called 0D deformation experiment (integration point test). The major purpose of this test is to show that local iterations adequately calculate stresses during the switch from the visco-elastic deformation into mode-I failure, as well as during the transition between mode-I and mode-II. Material properties are assumed to be homogeneous and magnitudes of the background strain rates components are selected such that three scenarios are generated, namely: i) constant volumetric strain rate is applied $\dot{\theta} \neq 0$ (volumetric extension test) (Fig. 5a), ii) constant deviatoric strain rate is applied $\dot{\varepsilon}_{\mathrm{II}} \neq 0$ (deviatoric shear test) (Fig. 5b), and iii) combination of both strain rates is applied (mixed strain test) (Fig. 5c). For simplicity we consider perfect plasticity case, i.e. we ignore softening and deactivate viscoplastic regularization. All three scenarios are initialized with zero stresses and subsequently, either pressure or deviatoric stress, or both, evolve until mode-I failure is activated. At later stages the stress evolution only occurs along the yield surface. For the constant volumetric strain rate case, the pressure simply stops changing when it reaches



**Table 1.** Material, discretization, and boundary condition parameters employed in presented models

| Parameter | Units | 0D Fig. 5a/5b | Regularization Fig. 6b/6d | Crust Fig. 7a/7b | Tensile Fig. 8 | Brittle-ductile Fig. 9 |
|---|---|---|---|---|---|---|
| $\rho$ | $[\text{kg m}^{-3}]$ | - | - | $3 \times 10^3$ | $3 \times 10^3$ | $3 \times 10^3$ |
| $B_\text{D}$ | $[\text{Pa}^{-1}\,\text{s}^{-1}]$ | $5 \times 10^{-21}$ | - | - | - | $5 \times 10^{-24}$ |
| $B_\text{N}$ | $[\text{Pa}^{-n}\,\text{s}^{-1}]$ | - | - | - | - | $8.8971 \times 10^{-25}$ |
| $E_\text{N}$ | $[\text{J mol}^{-1}]$ | - | - | - | - | $1.9 \times 10^5$ |
| $n$ | $[\ ]$ | - | - | - | - | $3.3$ |
| $G$ | $[\text{Pa}]$ | $10^{10}$ | $4 \times 10^{10}$ | $4 \times 10^{10}$ | $4 \times 10^{10}$ | $5 \times 10^{10}$ |
| $K$ | $[\text{Pa}]$ | $2 \times 10^{11}$ | $6.4 \times 10^{10}$ | $6.4 \times 10^{10}$ | $6.4 \times 10^{10}$ | $1.1 \times 10^{11}$ |
| $\varphi$ | $[°]$ | $30$ | $30$ | $30$ | $30$ | $30$ |
| $\psi$ | $[°]$ | $10$ | $0$ | $0$ | $0$ | $3$ |
| $c_\text{MC}^\text{init}$ | $[\text{Pa}]$ | $10^6$ | $2 \times 10^7$ | $2 \times 10^7$ | $2 \times 10^7$ | $2 \times 10^7$ |
| $c_\text{MC}^\text{min}$ | $[\text{Pa}]$ | - | $5 \times 10^6$ | $5 \times 10^6$ | $5 \times 10^6$ | $5 \times 10^6$ |
| $H_c$ | $[\text{Pa}]$ | - | $-10^8$ | $-10^8$ | $-10^8$ | $-5 \times 10^8$ |
| $p_\text{T}$ | $[\text{Pa}]$ | $-5 \times 10^5$ | $-10^6 / -10^7$ | $-10^6$ | $-10^6$ | $-10^6$ |
| $\eta^\text{vp}$ | $[\text{Pa s}]$ | - | $10^{19} / 5 \times 10^{18}$ | $10^{18}$ | $10^{18}$ | $10^{19}$ |
| L | $[\text{m}]$ | - | $1$ | $4 \times 10^4$ | $4 \times 10^4$ | $10^5$ |
| H | $[\text{m}]$ | - | $0.7$ | $7 \times 10^3$ | $7 \times 10^3$ | $2.5 \times 10^4$ |
| $A_\triangle$ | $[\text{m}^2]$ | - | $5 \times 10^{-6} - 3 \times 10^{-4}$ | $2.5 \times 10^3$ | $2.5 \times 10^3$ | $3.6 \times 10^3 - 2.5 \times 10^4$ |
| $\Delta t$ | $[\text{yr}]$ | $2$ | $50$ | $50$ | $1$ | $10^3$ |
| $\dot{\epsilon}_\text{xx}^\text{bg}$ | $[\text{s}^{-1}] \times 10^{-15}$ | $2.333/0$ | $6.338/-6.338$ | $15.844/-15.844$ | $7.922$ | $1$ |
| $\dot{\epsilon}_\text{yy}^\text{bg}$ | $[\text{s}^{-1}] \times 10^{-15}$ | $2.333/0$ | $0/6.338$ | - | - | - |
| $\dot{\epsilon}_\text{zz}^\text{bg}$ | $[\text{s}^{-1}] \times 10^{-15}$ | $2.333/0$ | $0$ | $0$ | $0$ | $0$ |
| $\dot{\epsilon}_\text{xy}^\text{bg}$ | $[\text{s}^{-1}] \times 10^{-14}$ | $0/7$ | - | - | - | - |

Note: L - domain length, H - domain height, $A_\triangle$ - target triangular element area, $\dot{\epsilon}_{ij}^\text{bg}$ - background strain rates

the tensile strength of the material (Fig. 5a, d). In the deviatoric and mixed cases the mode-I failure eventually switches to the mode-II when stress evolution path passes through the delimiter point between the yield surface segments (Fig. 5b, c, d). Due to the nonzero dilatation angle, the pressure continues to grow even after switching to the mode-II regime. This behavior is caused by a combination of the nonzero volumetric plastic strain and overall incompressibility constraint ($\dot{\theta} = 0$), which results

in elastic compression and associated pressure increase. In general the stress integration tests deliver the expected results for the considered loading scenarios.





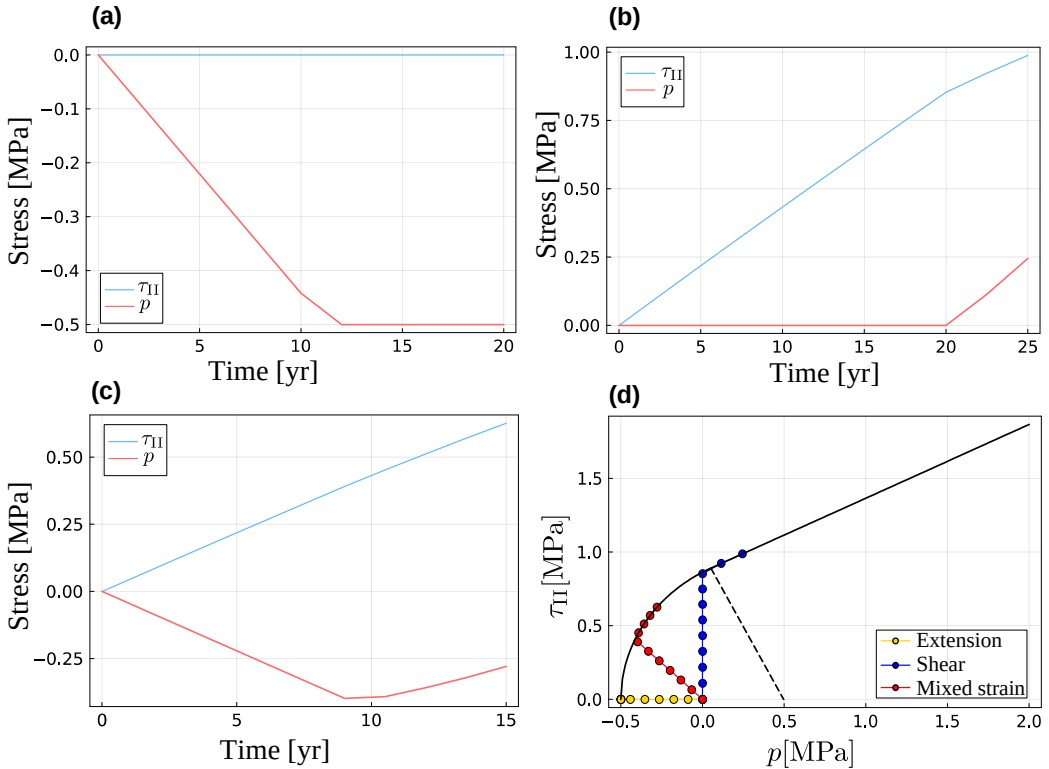

**Figure 5.** Temporal evolution of the effective deviatoric stress and pressure for the 0D stress integration tests. **(a)** Volumetric extension test $\dot{\theta} = 7 \times 10^{-15}\,\mathrm{s}^{-1}$ **(b)** Deviatoric shear test $\dot{\varepsilon}_{\mathrm{II}} = 7 \cdot 10^{-14}\,\mathrm{s}^{-1}$. **(c)** Mixed strain test (a combination of the volumetric extension and deviatoric shear used in a and b). **(d)** Stress evolution patterns in the meridional profile for all three tests. Both mode-I and transition between mode-I and mode-II are tested.

## 4.2 Viscoplastic regularization test

In the next example, we test the effect of viscoplastic regularization in a 2D setup that develops localized zones by either mode-I or mode-II plasticity. Two combinations of background strain rate loading is considered that correspond either to the uniaxial
restrained extension (Fig. 6a, b), or to the pure shear (Fig. 6c, d). For the extension case we plot the accumulated volumetric viscoplastic strain ($\chi$), which indicate loci of the tensile failure zones, whereas the deviatoric counterpart ($\kappa$) is plotted for the pure share case, which corresponds to the shear bands. The test cases are performed for different mesh resolutions. As expected, the non-regularized plasticity models demonstrates severe mesh dependency of both mode-I and mode-II plasticity cases, which results in localization zones that are close to one element in width and that have a maximum strain that depends on
the mesh resolution (Fig. 6a, c). With a suitably tuned viscoplastic regularization viscosity, on the other hand, mesh independent results are obtained (Fig. 6b, d), similar to earlier results for mode-II plasticity (Duretz et al., 2019, 2023). It should be noted that the Perzyna-type regularization we employ affects both deviatoric and volumetric components of viscoplastic strain rate





via the viscoplastic multiplier ($\dot{\lambda}$) (Eqs. 5, 9, and 11), and hence acts on mode-I and mode-II plasticity simultaneously. As previously reported by Duretz et al. (2019), one of the apparent advantages of the regularized models is its greatly improved convergence rate compared to the non-regularized models. In particular a few high resolution non-regularized models we tested here simply failed to converge due to the residual stagnation (Fig. 6a, c). In contrast to that, all the regularized models converged successfully within a prescribed number of iterations for both mode-I and mode-II cases (Fig. 6b, d). The typical convergence profiles of the nonlinear iterations are discussed in more detail in Sect. 4.6.

### 4.3 Strain localization in the brittle crust

Next, we consider the deformation of the upper crust in either extensional or compressional setting with an elasto-viscoplastic rheology. This problem represents a standard benchmark used in the geodynamics community to test various plasticity model implementations (see e.g. Kaus, 2010, and references therein). To facilitate the localization we use strain softening for cohesion parameter which is initiated by random accumulated deviatoric viscoplastic strain perturbations that are clustered around the central upper part of the domain. Subsequent deformation manifests itself in formation of multiple incipient localization zones (Fig. 7a), among which only few develop into fault zones at later stages (Fig. 7b). Under extension, mode-I plastic failure occurs close to the free surface, which is visible as vertically oriented high strain rate zones in Fig. 7a. With increasing depth and confining stress the localization switches again to mode-II plasticity. Under compression, the mode-I failure does not occur as mean stresses never become extensional. In general, it takes more time and horizontal strain to develop the localization zones in the compressional setting. In the considered setup the formation of the faults finishes at about 2.4 kyr under extension (Fig. 7b), whereas it still continues after about 7 kyr under compression (Fig. 7c). This can be explained by the increased compressive strength due to the increased dynamic pressure caused by the tectonic shortening. The orientation angles of the shear bands produced by the numerical models is the major observation parameter that is compared against the theoretical estimates for this benchmark problem. We refer to Kaus (2010) for the overview of the existing estimates as well as for the extensive discussion regarding the parameters that influence these orientations. Typically the result produced by the numerical models follow the theoretical estimates of Arthur et al. (1978), given by:

$$\alpha_{\mathrm{S}} = \frac{\pi}{4} \pm \frac{\varphi + \psi}{4}. \tag{69}$$

Here $\alpha_{\mathrm{S}}$ stands for the dip angles of the shear bands. The positive sign in this expression corresponds to the extensional setup, whereas the negative sign is attributed to compression. In general, normal faults are predicted to localize much steeper compared to the thrust faults. Detailed inspection of Fig. 7 reveals that Arthur's estimates are reproduced quite accurately in the presented simulations.

### 4.4 Tensile failure zone propagation

In this experiment, we consider propagation of a localized tensile failure zone induced by elevated fluid pressure. According to the theory of poroelasticity (Biot, 1941) the pressure variable in the momentum equation (Eq. 10) must be replaced with the




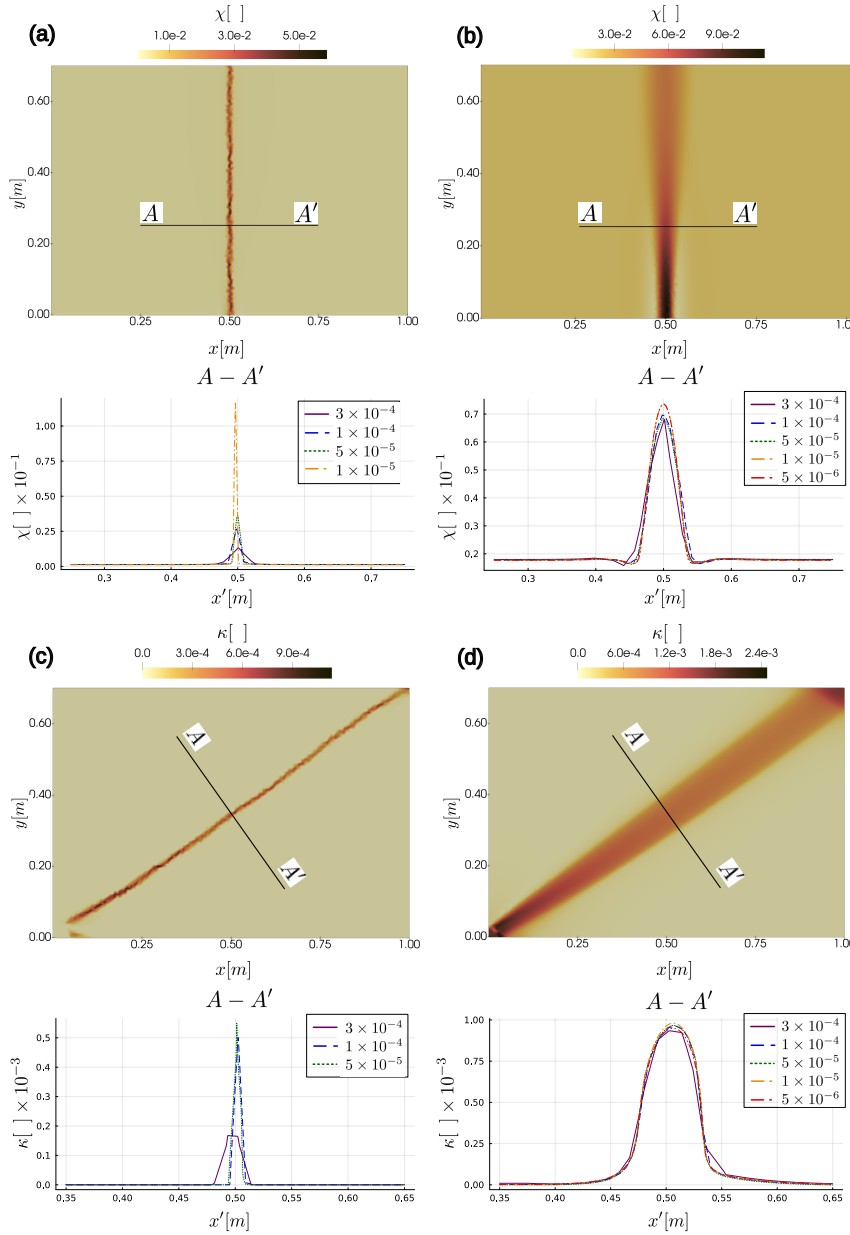

**Figure 6.** Results of viscoplastic regularization tests. **(a)** Tensile localization test without regularization after 10 kyr. **(b)** Tensile localization test with regularization ($\eta^{\mathrm{vp}} = 10^{19}$) after 100 kyr. **(c)** Shear localization test without regularization after 1 kyr. **(d)** Shear localization test with regularization ($\eta^{\mathrm{vp}} = 5 \times 10^{18}$) after 3 kyr. Curves on the cross-sections are labeled with target triangular element areas (mesh resolutions). Spatial distribution plots of $\chi$ and $\kappa$ correspond to the highest resolution tested for each case.



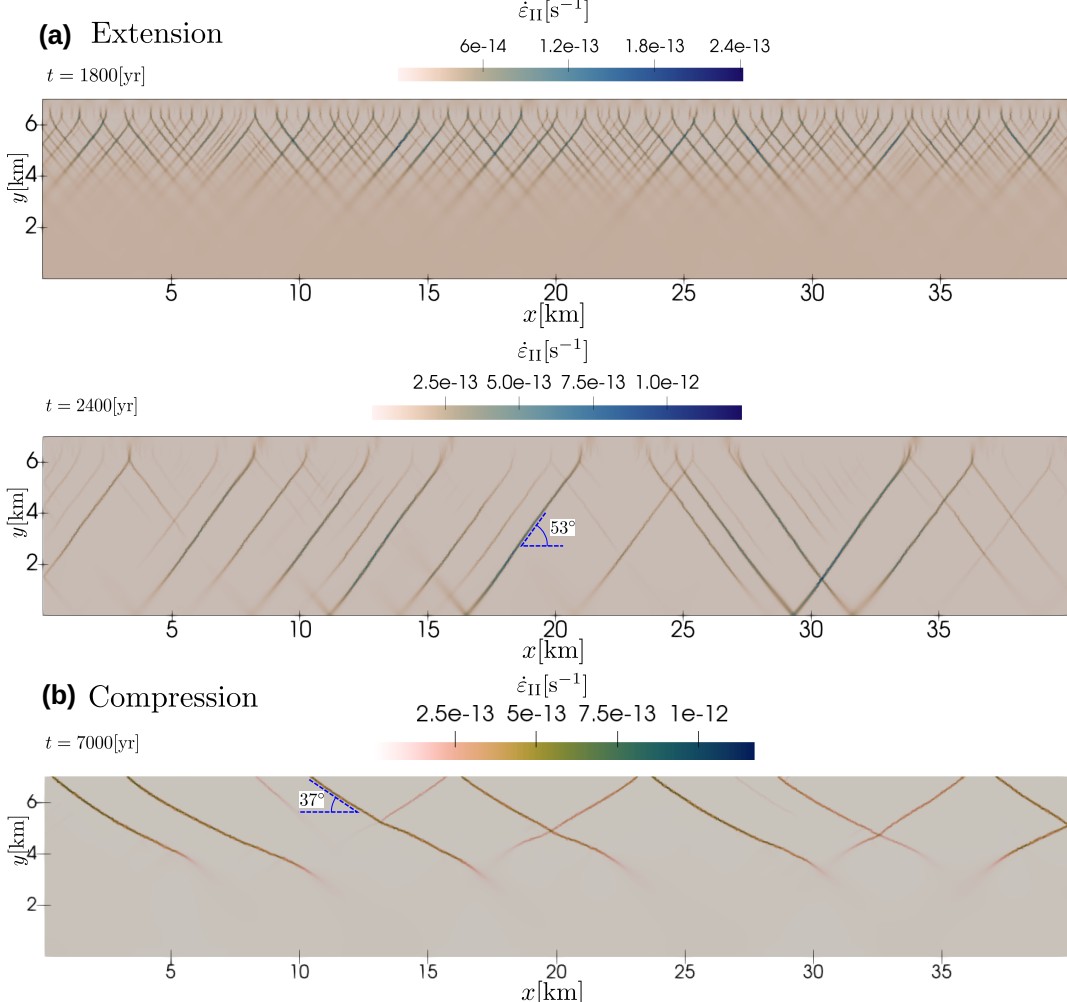

**Figure 7.** Spatial distribution of the deviatoric strain rate for the crustal strain localization problem. **(a)** Extension case after 1.8 kyr (incipient localization) and 2.4 kyr (localized faults). **(b)** Compression case after 7 kyr. Approximate shear band orientation angles are explicitly indicated. Note that the top of the domain is stress-free (representing the Earth's surface).

total pressure given by:

$$p_{\text{total}} = p + \alpha_{\text{B}}\, p_{\text{f}}, \tag{70}$$

where $p_{\text{f}}$ denotes the fluid pressure in the rock pores, $\alpha_{\text{B}}$ is the Biot-Willis coefficient, which depends on the bulk modulii of the rock matrix and solid grains (Biot and Willis, 1957), and $p$ is interpreted as the effective pressure that controls the deformation and failure of the porous rock. Apart from this single modification, the rest of the theory and notation in this paper remains unaltered. For the current test we set the Biot-Willis coefficient to one, which simply corresponds to Terzaghi's effective stress





principle (Terzaghi, 1925). For all other tests in this paper, we set the Biot-Willis coefficient to zero and thus ignore the effect
of fluid pressure.

The initial fluid pressure distribution is assumed to be homogeneous. We override it with a local perturbation of 180 MPa at
the center point of the bottom boundary. The perturbation is assumed to decay around the center point following a Gaussian
distribution with standard deviation of 200 m in the horizontal direction and 500 m in the vertical direction. The initial fluid
pressure perturbation is sufficiently large to initiate tensile failure. Subsequently the excessive fluid pressure is propagated to the
regions where the accumulated volumetric viscoplastic strain exceeds the prescribed limit, i.e.: $\chi > \chi_{\lim}$, which is assigned to a
relatively low value e.g. $10^{-4}$. This explicit procedure effectively mimics Darcy flow of the fluid into the enhanced permeability
zone induced by hydraulic fracturing, even when we do not model the actual underlying physical processes related to fluid flow,
in order to focus on the mechanical aspects of tensile failure.

The positive feedback between the plastic strain and the fluid pressure leads to a gradual propagation of the failure zone
in the vertical direction (Fig. 8a). The propagation velocity of the zone is increasing as the localization progresses towards
the surface. The fluid pressure in the zone remains almost the same as in the original perturbation due to the lack of viscous
dissipation (Fig. 8b), while the total pressure rapidly decreases due to decreasing overburden weight at shallower depths. Hence
the effective pressure becomes tensile and grows in magnitude, causing faster rock failure at the tip of the zone and therefore
faster propagation rates. The velocities at the boundaries of the domain are essentially controlled by the imposed background
strain rates. However, near the tip of the zone they reach values that are orders of magnitude larger than at the boundaries (Fig.
8c), suggesting that the dynamics of the system is mainly driven by the brittle failure and associated elastic unloading. The
animation of the time evolution of the horizontal velocity field is provided in the video supplement.

It should be noted that this setup is not intended to be interpreted as dyke propagation through the brittle crust even though
it resembles it rather closely. The reasons for this are numerous. We ignore thermal effects, do not represent realistic magma
viscosities and densities in the model, describe fluid flow by a simplified parametrization, etc. In reality, dikes and hydraulic
fractures form on much smaller length- and time-scales, compared to what is represented here. More realistic dyke propagation
models would require much finer numerical resolution.

## 4.5    Brittle-ductile transition

The tests so far focused on elasto-viscoplastic rheologies. Yet, our numerical formulation can also deal with linear and nonlinear
creep laws combined with elasto-viscoplastic failure models. To demonstrate that, our final test considers a similar crustal
extension setup we already introduced in Sect. 4.3. However this time we significantly increase the horizontal and vertical
spans of the domain to be $100 \times 25$ km. The domain is discretized with a variable grid resolution. The central upper part of the
domain with a size of $40 \times 7$ km has a target element area of 3600 m$^2$ (60 m average mesh size), which is just slightly greater
than in the crustal extension setup. The background element area is assigned to 25000 m$^2$ (500 m average mesh size). Within
a 20 km transition zone the element size is assumed to linearly grow from the refined value to a background value to prevent
sharp contrast in grid resolution. We assigned a background geothermal gradient to 20 $^\circ$C km$^{-1}$ and extend the crust in the
horizontal direction with a constant background strain rate. A combination of the selected background geothermal gradient, a



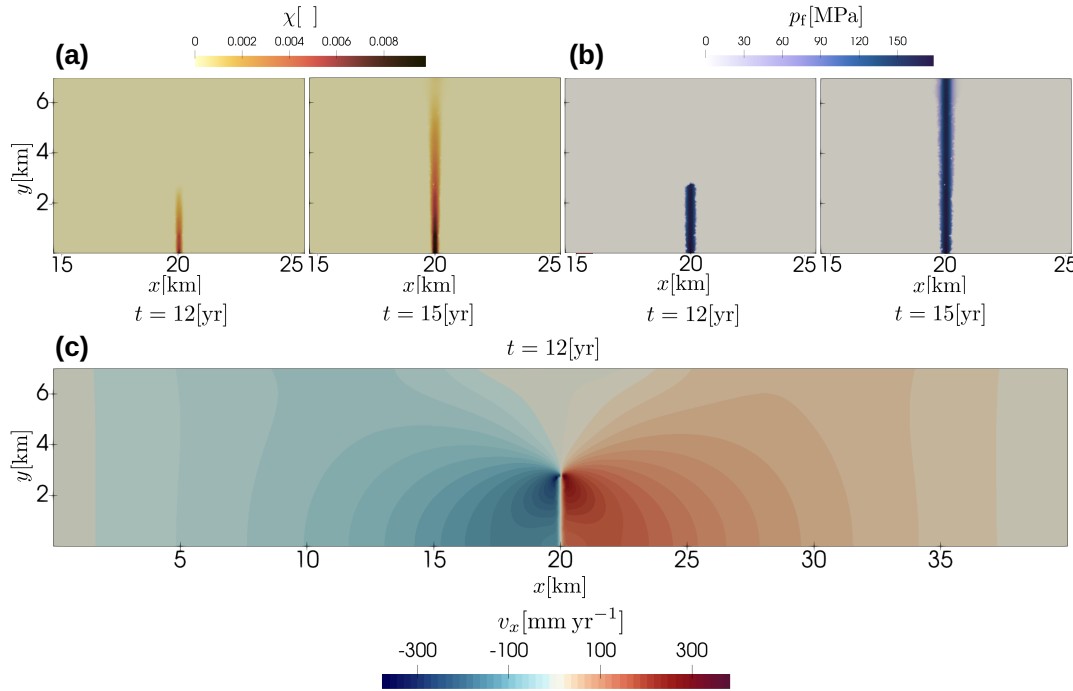

**Figure 8.** Results of tensile failure zone propagation problem. **(a)** Accumulated volumetric viscoplastic strain distribution after 12 yr (onset) and 15 yr (breakthrough). **(b)** Fluid pressure distribution after 12 yr and 15 yr. **(c)** Horizontal velocity distribution after 12 yr.

constant extension rate and a temperature- and stress-dependent dry upper crust quartzite rheology (Schmalholz et al., 2009)
ensures that the brittle-ductile transition occurs within the model domain, which is confirmed by a typical strength envelope of
the crust (Fig. 9c). In the brittle part of the crust we again obtain a combination of mode-I and mode-II plastic failure zones,
which develop in the central part of the domain (Fig. 9a, b). The localization is triggered by similar random perturbations of the
accumulated deviatoric viscoplastic strain as we used in the crustal setup (see Sect. 4.3 for the details). Peculiar features that
we obtain in our models are the vertical mode-I failure zones that occur close to the free surface where compressive strength is
the smallest due to limited overburden stress (Fig. 9b). It should be noted that these features are not reproducible by the models
that do not include a tensile yield surface (Popov and Sobolev, 2008; Kaus, 2010; Duretz et al., 2021; Jacquey and Cacace,
2020a, b). The animation of the time evolution of the accumulated deviatoric viscoplastic strain field is provided in the video
supplement.

## 4.6 Convergence of the global iterations

Global Newton-Raphson iteration, which we use in this paper, is a powerful technique to solve nonlinear systems of equations.
Nevertheless, it is well known that non-regularized plasticity models usually demonstrate very poor convergence rates, residual
stagnation, or even divergence caused by the unlimited residual growth. As previously reported by e.g. Duretz et al. (2019),



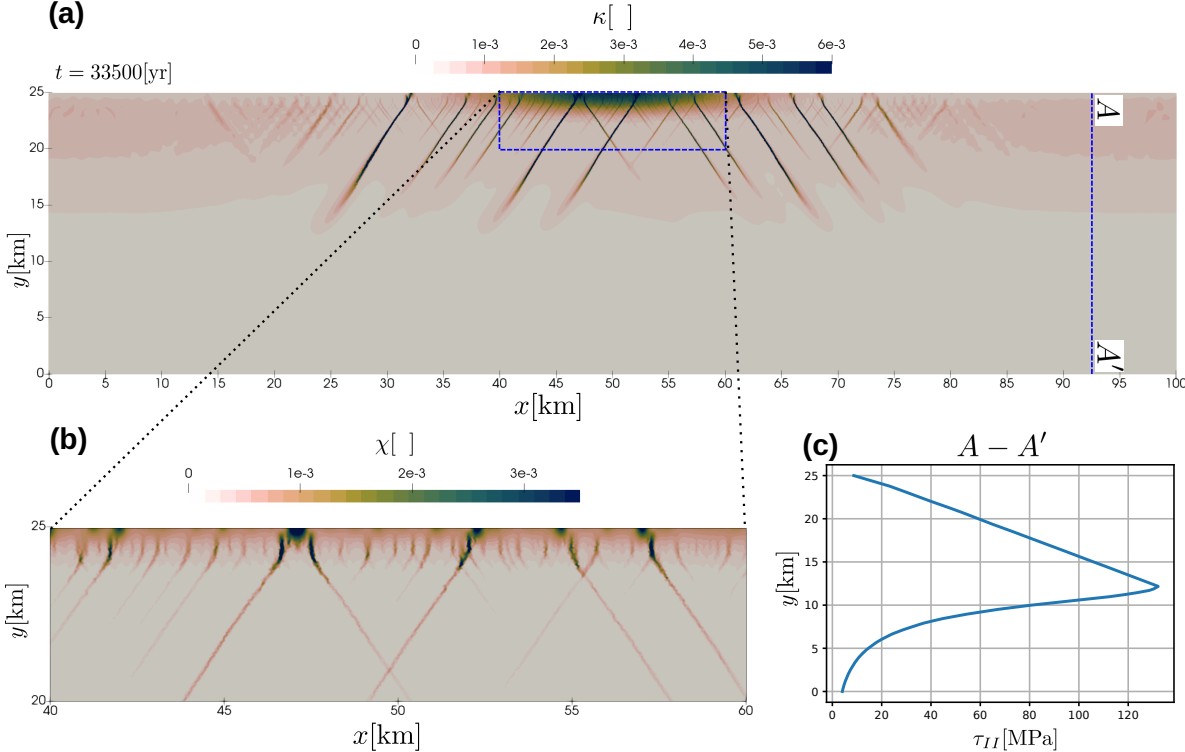

**Figure 9.** Results of brittle-ductile transition problem. **(a)** Accumulated deviatoric viscoplastic strain distribution after 33.5 kyr. **(b)** Accumulated volumetric viscoplastic strain distribution in a near-surface region. **(c)** Depth distribution of the effective deviatoric stress magnitude in the far field. The top of the domain is stress-free.

the positive side effect of the viscoplastic regularization is its significantly more robust convergence properties compared to the non-regularized cases. Here, we generally confirm this observation. For a typical crustal extension problem considered in Sect.4.3 only few (generally less than 10, and normally about 4-5) nonlinear iterations are necessary to achieve a relative tolerance of $10^{-5}$ (Fig. 10). In cases where an overwhelming amount of loading is applied during a single time step, which typically happens when a lot of fault zones are active simultaneously, the residual starts to stagnate and the nonlinear solve fails to converge within a prescribed number of iterations. In this situation, we use an adaptive time stepping, which simply halves the timestep and restarts the nonlinear solution procedure. After a prescribed number of successfully converged reduced times steps, the algorithm attempts to double the time step magnitude again to avoid excessively refined time discretization. It should finally be noted that the robustness of the nonlinear solution can be potentially further improved by a line search technique, similar to what we apply for the local stress update. In this paper, however, we did not explore this possibility for the global equilibrium iterations.



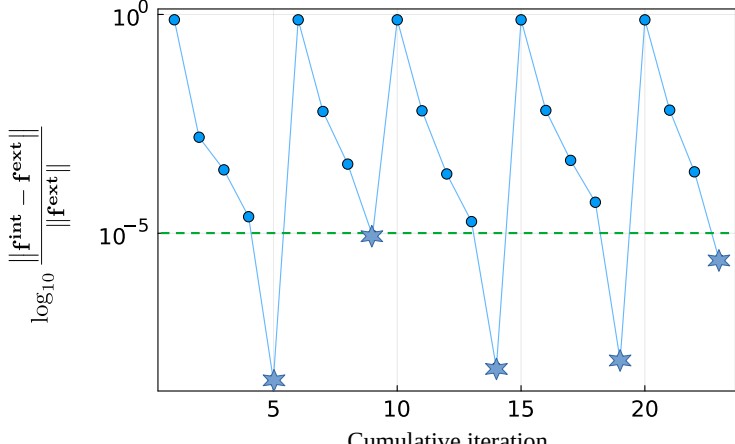

**Figure 10.** Snapshot of the typical nonlinear global convergence pattern for the crustal extension model (Sect. 4.3). Shown are the logarithm of the relative momentum residual norm vs. the cumulative iteration number. The partition of the momentum residual into internal and external forces is given by Eq.(67). Green dashed line represents the relative tolerance value used to terminate the iterations. Stars indicate converged residual norms of the time steps.

## 5 Conclusions

We proposed a relatively simple plasticity model that combines a linearized Drucker-Prager shear failure envelope with a circular tensile cap function in a way that ensures a globally smooth transition between these components. Similar models were elaborated before, however, here we additionally eliminate all singularities and discontinuities in the entire stress space for both the composite yield surface, and its corresponding flow potential. This important addition makes our plasticity model a suitable candidate for implementation in the context of Perzyna-type viscoplasticity, which is a key component to obtain

regularized solutions that are independent of the mesh resolution. The proposed plasticity model is characterized by a unique orientation of the flow potential gradient at any stress point, which in turn allows to formulate a relatively simple and robust local stress integration algorithm. Nevertheless, we found it necessary to use a line search algorithm to increase the stability of the local iterations and to achieve a practically acceptable level of robustness. Our rheological model also supports an arbitrary combination of viscoplasticity with linear and nonlinear viscous creep mechanisms, elasticity and strain softening. The amount

of dilatation in the shear failure envelope does not affect the formulation of the tensile cap surface, which implies that zero dilatation angle cases are explicitly supported.

     We implemented the proposed plasticity model in a new unstructured finite element code that uses LBB-stable conforming triangular Crouzier-Raviart elements to treat the near incompressibility constraint. Our mixed two-field formulation employs increments of displacements and pressure as primary variables. We found that treating the globally discretized pressure variable

as a trial visco-elastic pressure (trial pressure scheme) performs superior compared to treating it as a true spherical part of the Cauchy stress tensor (true pressure scheme). We also found that displacement increments are advantageous compared to



velocities as they help to control kinematical loading by changing the time step magnitude, while posing no restrictions on the type of rheology employed.

A number of 0D and 2D examples are provided that demonstrate the applicability of the proposed plasticity model to a set
of practically relevant cases that involve both mode-I and mode-II plasticity. These range from fluid-induced tensile failure zone propagation to strain localization in the visco-elasto-plastic crust. In general, it can be concluded that the algorithms and models presented in this work are quite robust and can be easily implemented in the framework of mixed finite element or staggered-grid finite difference formulations to simulate a wide range of geomechanical applications.

*Code availability.*

The current version of GeoTech2D code is available from GitHub website https://github.com/UniMainzGeo/GeoTech2D under the MIT license. The exact version of the code used in this work, the setup scripts to run the models, and the mesh generation scripts are archived on Zenodo under https://doi.org/10.5281/zenodo.15496843 (Popov and Kaus, 2025).

*Video supplement.*

On the GitHub page and Zenodo repository listed above, we provide a set of animations to illustrate the evolution of the
shear and tensile localization zones. The directory VIDEO contains the following animation files:

1. Tensile failure zone propagation problem (Sect. 4.4)

    (a) dyke_Pf_25.mp4 - fluid pressure

    (b) dyke_Vx_25.mp4 - horizontal velocity

2. Brittle-ductile transition problem (Sect. 4.5)

(a) ductile_EII.mp4 - effective deviatoric strain rate

    (b) ductile_aps.mp4 - accumulated deviatoric viscoplastic strain

**Appendix A: Stress update**

Applying the time discretization to the Jaumann stress rate (Eq. 6) we obtain:

$$\overset{\diamond}{\tau}_{ij} = \frac{\tau_{ij} - (\tau_{ij})_n}{\Delta t} + (\tau_{ik})_n \, \omega_{kj} - \omega_{ik} \, (\tau_{kj})_n. \tag{A1}$$

Taking the deviatoric projection of plastic flow potential gradient (Eq. 21) gives:

$$\mathrm{dev} \left( \frac{\partial Q}{\partial \sigma_{ij}} \right) = B_\tau \tau_{ij}. \tag{A2}$$





Upon substitution of both results in the deviatoric constitutive equation (Eq. 5) we finally get:

$$\dot{\varepsilon}_{ij} - \frac{1}{2G}\left[\frac{\tau_{ij} - (\tau_{ij})_n}{\Delta t} + (\tau_{ik})_n\,\omega_{kj} - \omega_{ik}\,(\tau_{kj})_n\right] - A_{\mathrm{D}}\,\tau_{ij} - A_{\mathrm{N}}\,(\tau_{\mathrm{II}})^{n-1}\,\tau_{ij} - \dot{\lambda}\,B_\tau\tau_{ij} = 0. \tag{A3}$$

Rearranging the results yields:

$$\dot{\varepsilon}^*_{ij} - A_{\mathrm{L}}\,\tau_{ij} - A_{\mathrm{N}}\,(\tau_{\mathrm{II}})^{n-1}\,\tau_{ij} - \dot{\lambda}\,B_\tau\tau_{ij} = 0, \tag{A4}$$

where the effective linear visco-elastic creep prefactor is defined as:

$$A_{\mathrm{L}} = \frac{1}{2G\Delta t} + A_{\mathrm{D}}, \tag{A5}$$

and the effective deviatoric strain rate and rotated deviatoric stresses from the previous time step, respectively, are given by:

$$\dot{\varepsilon}^*_{ij} = \dot{\varepsilon}_{ij} + \frac{\tau^*_{ij}}{2G\Delta t}, \quad \tau^*_{ij} = (\tau_{ij})_n + \Delta t\left[\omega_{ik}\,(\tau_{kj})_n - (\tau_{ik})_n\,\omega_{kj}\right]. \tag{A6}$$

From equation A4 it becomes obvious that the updated deviatoric stress and the effective strain rate are proportional to each other, which implies that all terms can be replaced with their corresponding scalar norms:

$$\dot{\varepsilon}^*_{\mathrm{II}} - A_{\mathrm{L}}\,\tau_{\mathrm{II}} - A_{\mathrm{N}}\,(\tau_{\mathrm{II}})^n - \dot{\lambda}\,A_\tau = 0 \tag{A7}$$

where the deviatoric plastic constant is computed as:

$$A_\tau = B_\tau\tau_{\mathrm{II}} = \begin{cases} \frac{1}{2}, & \tau_{\mathrm{II}}\,(p_{\mathrm{q}} - p_{\mathrm{d}}) \geq \tau_{\mathrm{d}}\,(p_{\mathrm{q}} - p) \\[2mm] \frac{b\,\tau_{\mathrm{II}}}{2\hat{R}_{\mathrm{q}}}, & \text{otherwise} \end{cases}. \tag{A8}$$

The stress derivatives of the deviatoric plastic constant are given by:

$$\left(\frac{\partial A_\tau}{\partial \tau_{\mathrm{II}}}, \frac{\partial A_\tau}{\partial p}\right) = \begin{cases} (0,0), & \tau_{\mathrm{II}}\,(p_{\mathrm{q}} - p_{\mathrm{d}}) \geq \tau_{\mathrm{d}}\,(p_{\mathrm{q}} - p) \\[2mm] \left(\frac{b(\hat{R}_{\mathrm{q}}^2 - \tau_{\mathrm{II}}^2)}{2\hat{R}_{\mathrm{q}}^3}, -\frac{b\tau_{\mathrm{II}}(p - p_{\mathrm{q}})}{2\hat{R}_{\mathrm{q}}^3}\right), & \text{otherwise} \end{cases}. \tag{A9}$$

For the continuity equation, we follow essentially the same steps as for the deviatoric equation. By discretizing the pressure time derivative we get:

$$\frac{Dp}{Dt} = \frac{p - p_n}{\Delta t}. \tag{A10}$$

The spherical part of the plastic flow potential gradient (Eq. 21) reads:

$$\mathrm{tr}\left(\frac{\partial Q}{\partial \sigma_{ij}}\right) = 3B_p. \tag{A11}$$

Substituting both terms in the continuity equation (Eq. 9) gives:

$$\dot{\theta} = -\frac{p - p_n}{K\Delta t} + 3\dot{\lambda}B_p. \tag{A12}$$





Assuming zero viscoplastic multiplier, the volumetric strain rate can be expressed in terms of the trial pressure as:

$$\dot{\theta} = -\frac{p^* - p_n}{K\Delta t}.$$ (A13)

Combining the right hand sides of both equations and rearranging the result, we finally obtain the local continuity residual:

$$\frac{p - p^*}{K\Delta t} - \dot{\lambda}\,A_p = 0,$$ (A14)

where the volumetric plastic constant is computed as:

$$A_p = 3B_p = \begin{cases} k_{\mathrm{q}}, & \tau_{\mathrm{II}}\,(p_{\mathrm{q}} - p_{\mathrm{d}}) \geq \tau_{\mathrm{d}}\,(p_{\mathrm{q}} - p) \\[3mm] -\frac{b(p - p_{\mathrm{q}})}{\hat{R}_{\mathrm{q}}}, & \text{otherwise} \end{cases}.$$ (A15)

The stress derivatives of the volumetric plastic constant are given by:

$$\left(\frac{\partial A_p}{\partial \tau_{\mathrm{II}}}, \frac{\partial A_p}{\partial p}\right) = \begin{cases} (0,0), & \tau_{\mathrm{II}}\,(p_{\mathrm{q}} - p_{\mathrm{d}}) \geq \tau_{\mathrm{d}}\,(p_{\mathrm{q}} - p) \\[3mm] \left(\frac{b\tau_{\mathrm{II}}(p - p_{\mathrm{q}})}{\hat{R}_{\mathrm{q}}^3}, -\frac{b\left(\hat{R}_{\mathrm{q}}^2 - (p - p_{\mathrm{q}})^2\right)}{\hat{R}_{\mathrm{q}}^3}\right), & \text{otherwise} \end{cases}.$$ (A16)

Finally, the Perzyna viscoplastic constitutive equation (Eq. 11) can be recast into a residual form:

$$\dot{\lambda} = \frac{\langle F \rangle}{\eta^{\mathrm{vp}}} \rightarrow \langle F \rangle - \dot{\lambda}\,\eta^{\mathrm{vp}} = 0.$$ (A17)

The stress derivatives of the composite yield surface are given by:

$$\left(\frac{\partial F}{\partial \tau_{\mathrm{II}}}, \frac{\partial F}{\partial p}\right) = \begin{cases} (1, -k), & \tau_{\mathrm{II}}\,(p_{\mathrm{y}} - p_{\mathrm{d}}) \geq \tau_{\mathrm{d}}\,(p_{\mathrm{y}} - p) \\[3mm] \left(\frac{a\tau_{\mathrm{II}}}{\hat{R}_{\mathrm{y}}}, \frac{a(p - p_{\mathrm{y}})}{\hat{R}_{\mathrm{y}}}\right), & \text{otherwise} \end{cases}.$$ (A18)

**Appendix B: Tangent operator**

To simplify the notation, we omit the star superscript indicating the effective deviatoric strain rates $(\dot{\varepsilon}_{ij}^*)$ in the following derivations. The linearization of the stress rotation terms, in general, depends on the employed advection algorithm. It can be relatively simple in case that rotation is performed in the integration points or control volumes. However, linearization of the

rotation terms performed on the material particles during the advection can be notoriously difficult. Since no general algorithm can be provided we assume that stress rotation terms do not contribute to the linearization, which is a fair assumption for geoscientific applications as rocks reach their ultimate yield stress and break when shear stresses do not exceed about 10% of the elastic shear modulus. At this stress level, stress rotation terms do not play a crucial role yet (see e.g., Thielmann et al., 2015). We therefore limit the derivations to the linearization of the material nonlinearity.





Directly differentiating the Newton update Eq. (44) with respect to the effective deviatoric strain rate ($\dot{\varepsilon}_{\mathrm{II}}$), and the trial pressure ($p^*$), assuming a unit step length ($\alpha = 1$), yields the following matrix:

$$
\mathbf{A} = \begin{bmatrix} \frac{\partial \tau_{\mathrm{II}}}{\partial \dot{\varepsilon}_{\mathrm{II}}} & \frac{\partial \tau_{\mathrm{II}}}{\partial p^*} \\[2mm] \frac{\partial p}{\partial \dot{\varepsilon}_{\mathrm{II}}} & \frac{\partial p}{\partial p^*} \\[2mm] \frac{\partial \dot{\lambda}}{\partial \dot{\varepsilon}_{\mathrm{II}}} & \frac{\partial \dot{\lambda}}{\partial p^*} \end{bmatrix} = -\mathbf{J}^{-1} \begin{bmatrix} 1 & 0 \\[2mm] 0 & -\frac{1}{K\Delta t} \\[2mm] 0 & 0 \end{bmatrix},
\tag{B1}
$$

which can be evaluated numerically after convergence of the local iterations. The coefficients of this matrix represent the numerical values of the correspondent derivatives. They can be directly used in the evaluation of the tangent operator.

Next, we differentiate the expression for the volumetric-deviatoric decomposition of the Cauchy stresses (Eq. 3) with respect to the strain rate tensor, which yields:

$$
\frac{\partial \sigma_{ij}}{\partial \dot{\epsilon}_{kl}} = \frac{\partial \tau_{ij}}{\partial \dot{\varepsilon}_{mn}} \frac{\partial \dot{\varepsilon}_{mn}}{\partial \dot{\epsilon}_{kl}} - \delta_{ij} \frac{\partial p}{\partial \dot{\varepsilon}_{\mathrm{II}}} \frac{\partial \dot{\varepsilon}_{\mathrm{II}}}{\partial \dot{\varepsilon}_{mn}} \frac{\partial \dot{\varepsilon}_{mn}}{\partial \dot{\epsilon}_{kl}}.
\tag{B2}
$$

The derivative of the deviatoric strain rate (Eq. 4) with respect to the total strain rate gives the unit deviatoric projection tensor, which is given by:

$$
\frac{\partial \dot{\varepsilon}_{ij}}{\partial \dot{\epsilon}_{kl}} = I^D_{ijkl} = I_{ijkl} - \frac{1}{3}\delta_{ij}\delta_{kl}, \quad I_{ijkl} = \frac{1}{2}\left(\delta_{ik}\delta_{jl} + \delta_{il}\delta_{jk}\right).
\tag{B3}
$$

By differentiating the norm of the effective deviatoric tensor with respect to the tensor itself, we obtain the scaled normalized deviatoric direction tensor given by the Eq. (49):

$$
\frac{\partial \dot{\varepsilon}_{\mathrm{II}}}{\partial \dot{\varepsilon}_{kl}} = \frac{1}{2}n_{kl}.
\tag{B4}
$$

The derivative of the deviatoric stress update expression (Eq. 48) reads:

$$
\frac{\partial \tau_{ij}}{\partial \dot{\varepsilon}_{kl}} = \frac{\partial \tau_{\mathrm{II}}}{\partial \dot{\varepsilon}_{\mathrm{II}}} \frac{\partial \dot{\varepsilon}_{\mathrm{II}}}{\partial \dot{\varepsilon}_{kl}} n_{ij} + \tau_{\mathrm{II}} \frac{\partial n_{ij}}{\partial \dot{\varepsilon}_{kl}}.
\tag{B5}
$$

Finally, differentiating the expression for the deviatoric direction tensor (Eq. 49) and using Eq. (B4)gives:

$$
\frac{\partial n_{ij}}{\partial \dot{\varepsilon}_{kl}} = \frac{1}{\dot{\varepsilon}_{\mathrm{II}}} \frac{\partial \dot{\varepsilon}_{ij}}{\partial \dot{\varepsilon}_{kl}} - \frac{\dot{\varepsilon}_{ij}}{\dot{\varepsilon}^2_{\mathrm{II}}} \frac{\partial \dot{\varepsilon}_{\mathrm{II}}}{\partial \dot{\varepsilon}_{kl}} = \frac{1}{\dot{\varepsilon}_{\mathrm{II}}} \left( I_{ijkl} - \frac{1}{2}n_{ij}n_{kl} \right).
\tag{B6}
$$

Substituting Eqs. (B3) − (B6) into Eq. (B2), making use of the matrix coefficients from Eq. (B1), and simplifying the resulting expression we get:

$$
\frac{\partial \sigma_{ij}}{\partial \dot{\epsilon}_{kl}} = 2\,\eta_{\mathrm{eff}} I^D_{ijkl} + \beta_1 n_{ij}n_{kl} + \beta_2 \delta_{ij}n_{kl},
\tag{B7}
$$

where the effective stiffness constants are given by:

$$
\eta_{\mathrm{eff}} = \frac{\tau_{\mathrm{II}}}{2\,\dot{\varepsilon}_{\mathrm{II}}}, \quad \beta_1 = \frac{1}{2}A_{11} - \eta_{\mathrm{eff}}, \quad \beta_2 = -\frac{1}{2}A_{21}.
\tag{B8}
$$



The derivative of the Cauchy stress with respect to the global pressure reads:

$$\frac{\partial \sigma_{ij}}{\partial p^*} = \frac{\partial \tau_{ij}}{\partial p^*} - \delta_{ij} \frac{\partial p}{\partial p^*}. \tag{B9}$$

Differentiating the deviatoric stress update expression (Eq. 48) and keeping in mind that the deviatoric direction tensor does not depend on pressure, yields:

$$\frac{\partial \tau_{ij}}{\partial p^*} = \frac{\partial \tau_{\mathrm{II}}}{\partial p^*} n_{ij}. \tag{B10}$$

After substituting Eq. (B10) into Eq. (B9) and using the coefficients from Eq. (B1) we obtain:

$$\frac{\partial \sigma_{ij}}{\partial p^*} = \beta_3 n_{ij} + \beta_4 \delta_{ij}, \tag{B11}$$

where the effective stiffness constants are given by:

$$\beta_3 = A_{12}, \quad \beta_4 = -A_{22}. \tag{B12}$$

For the visco-elastic case, $A_{11}$ becomes the only non-trivial matrix coefficient in the Eq. (B1). A quick inspection of Eq.(47) along with Eq. (B1) reveals that it can be directly evaluated as:

$$A_{11} = \left( A_L + n\, A_N \tau_{II}^{n-1} \right)^{-1}. \tag{B13}$$

*Author contributions.* AP, NB and BK elaborated the algorithmic concept; AP and NB implemented the codes, conducted the numerical experiments, visualized the results, and prepared the manuscript; BK prepared the animations. All authors reviewed and edited the manuscript.

*Competing interests.* One of the (co-)authors is a member of the editorial board of *Geoscientific Model Development*.

*Acknowledgements.* This research was supported by the European Research Council through ERC Consolidator Grant #771143 (MAGMA). This paper is partly based on the PhD thesis of Nicolas Berlie.



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
