# Peer review of "A dilatant visco-elasto-viscoplasticity model with globally continuous tensile cap: stable two-field mixed formulation"

_EGUsphere, 2025_

## Author Comment (AC1)

**RESPONSE TO REFEREES**

Blue font color is used to indicate our replies.

The text added to manuscript is highlighted in yellow.

The text removed from manuscript is highlighted in red.

We would like to thank both anonymous referees for their constructive comments as well as the topical editor for handling the manuscript. Below we address all the critical statements of the referees.

**Anonymous referee #1**

This is a very solid and interesting manuscript, which addresses the proper modelling of the transition between brittle and ductile failure. Publication is recommended, but the authors may wish to pay attention to the following issues in a revised version:

We appreciate the efforts of anonymous referee #1 and address all comments below.

* While viscosity (mostly) remedies mesh dependence, it should be pointed out that it is a weak regularisation technique in quasi-static loading conditions, as considered here. Only under dynamic loading conditions full regularisation of the governing equations can be proven, see De Borst and Duretz (2020). This should be discussed briefly in the paper in order to avoid misconceptions in the community.

We agree that the discussion about viscoplastic regularization for quasi-static cases was missing. This issue was also raised by anonymous referee #2 (see second comment of anonymous referee #2). Combining both constructive suggestions regarding this issue we have elaborated the following text in the introduction:

To avoid potential misconceptions, it should be noted that despite a satisfactory practical performance of viscoplastic regularization, a rigorous theoretical framework proving its effectiveness in quasi-static cases, such as considered in this work, does not currently exist (e.g., de Borst and Duretz, 2020).

… and later on …

Despite the lack of a solid theoretical background, we still advocate the use of viscoplastic regularization motivated by its successful application to multi-dimensional quasi-static cases, which can be traced back to the earlier works of Zienkiewicz and Cormeau (1972) and Zienkiewicz and Cormeau (1974) and which is fully supported by more recent publications we cite above, as well as the results shown here.

* Work on combining mode-I and mode-II plasticity has been pursued before in the context of modelling of concrete, albeit for plane stress rather than for plane strain or 3D conditions as is the

focus of the current paper, see e.g. Feenstra and de Borst. Int. J. Solids Structures (1996) 33, 707 - 730. It would be advisable to put the present contribution also in that context.

Yes, this is a relevant example. We appreciate referee for pointing it out. We have included this paper in the reference list and added the following text to the introduction:

An example of a plasticity model that handles a combination of tensile and shear failure of concrete in plane stress formulation can be found in Feenstra and de Borst (1996).

* The discussion on volumetric locking is confusing and perhaps misleading. It is suggested that volumetric locking occurs for isochoric plastic deformations and it is not (explicitly) pointed out that this phenomenon also occurs for dilatant or contractant plastic flow. Indeed, in all cases a kinematic constraint is imposed at failure, i.e. when the elastic deformations vanish, see De Borst and Groen, Int. J. Num. Meth. Engng (1995) 38, 2887 - 2906.

Indeed, we did not intend to say that volumetric locking is limited to volume-preserving plastic deformation. We have removed the following (confusing) text from the introduction:

Most plasticity models for rocks are pressure-dependent. Whereas mode-I plasticity models involve dilation or volumetric deformation, experiments demonstrate that mode-II faults usually involve only a bit of dilation during the initiation stages, but are dilation-free or incompressible after some deformation (e.g., Vermeer and de Borst, 1984). Thus, numerical models that incorporate general plasticity models for rocks need to have accurate pressure fields, and should be able to deal with both compressible and incompressible deformation.

This is of particular importance if the deformation mechanism is sensitive to pressure (as is the case for many plasticity models).

The following text was added instead, which hopefully clarifies the issue:

… which is well-known to require a stable discretization to avoid numerical artifacts, including volumetric and shear locking, pressure oscillations, and hourglass modes  (e.g., Zienkiewicz and Taylor, 2000). Moreover, the same numerical problems occur when plastic deformation dominates irrespective whether the flow is incompressible or dilatant/contractant, since volumetric plastic deformation can be effectively viewed as a form of kinematical constraint, which is similar to incompressibility (e.g., de Borst and Groen, 1995).

**Anonymous referee #2**

This is a robust manuscript looking at the addition of a tensile cap for modelling rock failure in a geodynamic context. The manuscript is of excellent quality and should eventually be published to benefit the community. There is however a point where the manuscript could be improved, regarding viscoplasticity and strain regularization.

We appreciate the efforts of anonymous referee #2 and address all comments below.

- The manuscript presents Perzyna's viscoplasticity as a way to regularize the ill-posed problem of strain localization. While this is valid way to consider viscoplasticity, there is no mention in the manuscript that viscoplasticity could also capture a physical behaviour of rate-dependent plastic deformation (which is not uncommon).

We agree and have added the following statement to the introduction:

It should also be mentioned that viscoplasticity extends beyond the regularization framework and may represent a true physical deformation mechanism that requires laboratory experiments to calibrate its material parameters.

- The authors claim that introducing viscoplasticity in a quasi-static approximation is sufficient to regularize the strain localization problem. I am not aware of any studies proving this statement. A number of studies analyzed the role of viscoplasticity for strain regularization and, to my knowledge, they all included the dynamic terms.

We agree with this critical statement. The discussion regarding the regularization of quasi-static cases was missing. Please refer to our reply on the first comment of anonymous referee #1, which raises the same issue.

- Furthermore, the authors should consider discussing other studies about viscoplasticity and strain regularization, some of which are more critical about the role of viscoplasticity than the cited studies in the manuscript (Jacquey et al. (2021) and Stathas and Stefanou (2022) as examples).

References :

Stathas, A. and Stefanou, I. The role of viscous regularization in dynamical problems, strain localization and mesh dependency. Computer Methods in applied Mechanics and Engineering, 388, 113185, 2022.

Jacquey, A. B.,  Rattez, H. and Veveakis, M. Strain localization regularization and patterns formation in rate-dependent plastic materials with multiphysics coupling. Journal of the Mechanics and Physics of Solids, 152, 104422, 2021.

We acknowledge that there is no general agreement in the literature at this stage about the success of viscoplastic regularization even for the dynamic cases. We would also like to point out, though, that the tests shown in previous work (such as in de Borst and Duretz, 2020), as well as in the current manuscript do seem to suggest that localization becomes mesh insensitive. Clearly more work is required to better understand these discrepancies between theoretical and practical studies. We have included both suggested publications in our discussion on this topic and added the following text to the introduction:

Moreover, even for dynamic cases, questions have recently been raised on the regularization potential of viscoplasticity, with some suggesting that it only works conditionally (e.g., Jacquey et al., 2021), whereas others suggest that it is unable to regularize a 1D simple shear setup (e.g., Stathas and Stefanou, 2022).